# Boosting the interfacial superionic conduction of halide solid electrolytes for all-solid-state batteries

Hiram Kwak[1,6], Jae-Seung Kim[2,6], Daseul Han[3,6], Jong Seok Kim[1], Juhyoun Park[1], Gihan Kwon [4], Seong-Min Bak [4,5], Unseon Heo[3], Changhyun Park [2], Hyun-Wook Lee [2], Kyung-Wan Nam [3] ✉, Dong-Hwa Seo [2] ✉ & Yoon Seok Jung [1] ✉

Designing highly conductive and (electro)chemical stable inorganic solid electrolytes using cost-effective materials is crucial for developing all-solid-state batteries. Here, we report halide nanocomposite solid electrolytes (HNSEs) $ZrO_2$(-ACl)-$A_2ZrCl_6$ (A = Li or Na) that demonstrate improved ionic conductivities at 30 °C, from 0.40 to 1.3 mS cm$^{-1}$ and from 0.011 to 0.11 mS cm$^{-1}$ for Li$^+$ and Na$^+$, respectively, compared to $A_2ZrCl_6$, and improved compatibility with sulfide solid electrolytes. The mechanochemical method employing $Li_2O$ for the HNSEs synthesis enables the formation of nanostructured networks that promote interfacial superionic conduction. Via density functional theory calculations combined with synchrotron X-ray and $^6$Li nuclear magnetic resonance measurements and analyses, we demonstrate that interfacial oxygen-substituted compounds are responsible for the boosted interfacial conduction mechanism. Compared to state-of-the-art $Li_2ZrCl_6$, the fluorinated $ZrO_2$−$2Li_2ZrCl_5F$ HNSE shows improved high-voltage stability and interfacial compatibility with $Li_6PS_5Cl$ and layered lithium transition metal oxide-based positive electrodes without detrimentally affecting Li$^+$ conductivity. We also report the assembly and testing of a Li-In|| $LiNi_{0.88}Co_{0.11}Mn_{0.01}O_2$ all-solid-state lab-scale cell operating at 30 °C and 70 MPa and capable of delivering a specific discharge of 115 mAh g$^{-1}$ after almost 2000 cycles at 400 mA g$^{-1}$.

Lithium-ion batteries are ubiquitous in electronic devices, and now their use is expanding to electric vehicles[1–3]. However, concerns about their safety and limited energy density motivate the development of all-solid-state batteries (ASSBs) exploiting nonflammable inorganic superionic conductors to enable alternative electrode materials such as Li metal[2–12].

Among various inorganic solid electrolyte (SE) candidates, sulfides, such as argyrodite $Li_{6-y}PS_{5-y}X_{1+y}$ (X = Cl, Br; y = 0.0−0.5; Li$^+$ conductivities range 1−10 mS cm$^{-1}$ at 25 °C)[13], have the advantages of high ionic conductivities reaching those of standard non-aqueous liquid electrolyte solutions and deformability, which allows for practical cold-pressing fabrication of ASSB cells[2,4]. However, due to their low

[1]Department of Chemical and Biomolecular Engineering, Yonsei University, Seoul 03722, South Korea. [2]School of Energy and Chemical Engineering, Ulsan National Institute of Science and Technology (UNIST), Ulsan 44919, South Korea. [3]Department of Energy and Materials Engineering, Dongguk University, Seoul 04620, South Korea. [4]National Synchrotron Light Source II, Brookhaven National Laboratory, Upton, NY 11973, USA. [5]Department of Materials Science and Engineering, Yonsei University, 03722 Seoul, South Korea. [6]These authors contributed equally: Hiram Kwak, Jae-Seung Kim, Daseul Han. ✉e-mail: knam@dongguk.edu; dseo@unist.ac.kr; yoonsjung@yonsei.ac.kr

intrinsic electrochemical oxidative limits (~2.6 V vs. Li/Li$^+$), uncoated 4 V class layered Ni-rich LiMO$_2$ (M = Ni, Co, Mn, or Al mixture) positive electrode active materials with sulfide SEs exhibit unsatisfactory performance[14–16]. Compared to sulfides, the other major class of SEs, oxides, such as Li$_7$La$_3$Zr$_2$O$_{12}$ (max. 1.8 mS cm$^{-1}$ at 27 °C)[5,17], have good (electro)chemical oxidation stabilities but are brittle, which makes it challenging to fabricate ASSBs without hot-sintering or hybridization with non-aqueous liquid electrolyte solutions[18,19].

Recently, halide SEs have emerged as a strong contender because they have a balance and combination of the advantages of sulfides and oxides, i.e., mechanical sinterability with good (electro)chemical stability[20,21]. In 2018, Asano et al. reported that mechanochemically prepared trigonal Li$_3$YCl$_6$ exhibited moderate Li$^+$ conductivity of 0.51 mS cm$^{-1}$ at 25 °C and good performance in ASSB cells, even when uncoated LiCoO$_2$-based positive electrode was used in combination with a Li-In negative electrode[20]. These results boosted research to develop halide SEs, such as Li$_3$InCl$_6$ (1.5 mS cm$^{-1}$ at 25 °C), Li$_3$ScCl$_6$ (3 mS cm$^{-1}$ at 25 °C), Li$_2$Sc$_{2/3}$Cl$_4$ (1.5 mS cm$^{-1}$ at 30 °C), Li$_2$ZrCl$_6$ (0.40 mS cm$^{-1}$ at 30 °C), and Li$_3$YbCl$_6$ (0.19 mS cm$^{-1}$ at 30 °C)[22–26]. Similar to other types of SE materials[2,4,6,13,27–29], the conventional strategy of compositional tuning, e.g., aliovalent substitution, to control the charge carrier concentration and/or structural framework was applied to enhance the ionic conductivity of halide SEs:[22,26,30,31] Li$_{3-x}$M$_{1-x}$Zr$_x$Cl$_6$ (M = Y, Er, max. 1.4 mS cm$^{-1}$ at 25 °C), Li$_{3-x}$M$_{1-x}$Zr$_x$Cl$_6$ (M = In, Sc, max. 2.1 mS cm$^{-1}$ at 30 °C), Li$_{2+x}$Zr$_{1-x}$M$_x$Cl$_6$ (M = Fe, Cr, V, max. 1.0 mS cm$^{-1}$ at 30 °C), and Li$_{3-x}$Yb$_{1-x}$M$_x$Cl$_6$ (M = Zr, Hf, max. 1.5 mS cm$^{-1}$ at 30 °C). Na$^+$ halide analogues such as Na$_{3-x}$Er$_{1-x}$Zr$_x$Cl$_6$ and Na$_{3-y}$Y$_{1-x}$Zr$_x$Cl$_6$ were also developed via aliovalent substitution, but their conductivities were as low as 0.040 mS cm$^{-1}$ at 25 °C and 0.066 mS cm$^{-1}$ at 20 °C, respectively (vs. 0.018 mS cm$^{-1}$ of Na$_2$ZrCl$_6$ at 30 °C)[32–34]. In addition, structural disorders, such as M (M = Y, Er) and/or Li$^+$ site disorder and stacking faults, varied depending on the preparation protocol and were identified as the key factors for enhancing the ionic conductivity of halide SEs[26,30,33,35].

In terms of practical applications, most halide SEs exploit scarce and expensive central metals, such as Y, Sc, and In, with the sole exception of Zr (Supplementary Fig. 1)[21,26,36]. Recent theoretical and experimental studies reported that the central metal cation and halide anion governed the electrochemical stability of halide SEs[37]. In particular, F-substitution in chloride SEs effectively pushes the electrochemical oxidative limit further[37] but at the expense of lower ionic conductivities[37,38]. In contrast to their good electrochemical oxidation stability, halide SEs suffer from poor cathodic stability associated with the reduction of central metal cations[14,24,38,39]. In this regard, an ASSB design, wherein the halide and sulfide SEs function synergistically as the catholyte and SE layer that is placed in-between negative and positive electrodes (hereafter, referred to as SE layer), respectively, is reasonable for practical ASSBs[7,20–26,30–32,38,40]. However, the compatibility issue of halide/sulfide has been overlooked thus far[41].

Since 1973, when Liang et al. discovered that the ionic conductivity of LiI improved from 10$^{-7}$ to 10$^{-5}$ S cm$^{-1}$ at 25 °C with the addition of Al$_2$O$_3$, the ionic conduction enhancement in heterostructured systems has been an intriguing but debatable question for various material systems, such as CaF$_2$/BaF$_2$ multilayered films, LiF/silica films, LiBH$_4$/Al$_2$O$_3$, and polymer electrolytes with inorganic fillers[42–48]. Because there is no clear understanding of such behaviour, the design principle for interfacial conduction enhancement has not been established. Therefore, it is critical to precisely understand the interfacial conduction mechanism to utilize the superionic conduction effect as a general material design principle. Specifically, application of the interfacial conduction strategy is not reported yet for any superionic conductor with ionic conductivity ≥1 mS cm$^{-1}$ at 25 °C, which is the minimum conductivity for practical ASSBs[49].

In this work, we report the mechanochemical preparation of Li$^+$- and Na$^+$-conducting halide nanocomposite SEs (HNSEs, e.g.,

ZrO$_2$-AX-A$_2$ZrX$_6$, A = Li or Na, X = Cl, F), that exhibit enhancements in not only ionic conductivities via interfacial superionic conduction (Li$_2$ZrCl$_6$: from 0.40 to 1.3 mS cm$^{-1}$ at 30 °C, Na$_2$ZrCl$_6$: 0.011 to 0.11 mS cm$^{-1}$ at 30 °C, hereafter, all reported conductivity values, in the absence of temperature information, are to be understood as having been obtained at 30 °C) but also compatibility with sulfide SEs. Density functional theory (DFT) calculations revealed the underlying interfacial superionic behaviour by establishing the interfacial conduction principles of HNSEs. These were experimentally probed by combined synchrotron-based X-ray and $^6$Li magic-angle spinning–nuclear magnetic resonance (MAS-NMR) measurements and analyses. In addition, the HNSE strategy was applied to F-substituted Li$_2$ZrCl$_6$, which offset the degradation in ionic conductivity and resolved the incompatibility issue with sulfide Li$_6$PS$_5$Cl (LPSCl) at elevated temperature. These HNSEs enabled good electrochemical energy storage performances of lab-scale cells with Li-In negative electrodes and LiCoO$_2$ (LCO) or single-crystalline LiNi$_{0.88}$Co$_{0.11}$Mn$_{0.01}$O$_2$ (S-NCM88) positive electrodes in terms of LPSCl compatibility at 60 °C, high-voltage stability, fast charging, and long-term cycle life.

## Results and discussion
### Synthesis and characterization of HNSEs

HNSEs were prepared by the mechanochemical reaction of LiCl (or NaCl) and ZrCl$_4$ with Li$_2$O (or Na$_2$O) (Fig. 1a). Li$_2$O acts as an oxygen source and reacts with ZrCl$_4$ to form ZrO$_2$ nanoparticles[50], and the residual ZrCl$_4$ and LiCl react to produce Li$_2$ZrCl$_6$. Based on DFT calculations, the reaction to generate ZrO$_2$ and LiCl from ZrCl$_4$ and Li$_2$O has a strong driving force (Supplementary Table 1, Supplementary Equation 1, ΔE = −4.736 eV). Furthermore, there is another spontaneous reaction from reactants (Li$_2$O, ZrCl$_4$, LiCl) to products (Li$_2$ZrCl$_6$, ZrO$_2$) when the molar ratios are stoichiometrically matched (Supplementary Table 1, Supplementary Equation 2, ΔE = −5.000 eV). Moreover, Li$_2$ZrCl$_6$ is the only stable phase in the ZrO$_2$-ZrCl$_4$-LiCl ternary region (Supplementary Fig. 2 and Supplementary Note 1). Consistently, the synchrotron X-ray diffraction (XRD) and pair distribution function (PDF) results for a precursor mixture of Li$_2$O and ZrCl$_4$ (2:3 molar ratio) during mechanochemical milling (Supplementary Figs. 3 and 4) confirm the mechanochemical synthesis of ZrO$_2$−2Li$_2$ZrCl$_6$ HNSEs with a negligible amount of precursors or impurities when the milling time is ≥ ≈20 h (Supplementary Note 2). Thus, the products of the mechanochemical reaction of LiCl and ZrCl$_4$ with Li$_2$O are comprised of Li$_2$ZrCl$_6$, ZrO$_2$, and LiCl, and their fractions are determined from the stoichiometric ratio of the precursors. We extensively characterized the ZrO$_2$−2Li$_2$ZrCl$_6$ HNSE sample as it is a binary system and exhibited a much higher Li$^+$ conductivity of 1.1 mS cm$^{-1}$ than Li$_2$ZrCl$_6$ (0.40 mS cm$^{-1}$), despite the 7.86 vol.% of ionically insulating ZrO$_2$ (based on the chemical formula of ZrO$_2$−2Li$_2$ZrCl$_6$). For comparison, a control sample was prepared by mechanochemical milling commercially available ZrO$_2$ nanoparticles (~20 nm) with Li$_2$ZrCl$_6$, referred to as nZrO$_2$−2Li$_2$ZrCl$_6$.

The XRD pattern of ZrO$_2$−2Li$_2$ZrCl$_6$ is compared with those of Li$_2$ZrCl$_6$ and nZrO$_2$−2Li$_2$ZrCl$_6$ in Fig. 1b. The main reflections for ZrO$_2$−2Li$_2$ZrCl$_6$ matched those of Li$_2$ZrCl$_6$ with hexagonal close-packed (hcp) trigonal structure (space group $P\bar{3}m1$), and their breadth indicated low crystallinity[20,26]. However, the XRD signals of ZrO$_2$ were not observed, suggesting nanosized grains with poor crystallinity. The local structures of the poorly crystalline ZrO$_2$−2Li$_2$ZrCl$_6$ HNSE were characterized by X-ray absorption spectroscopy (XAS) and PDF measurements, and the corresponding results are compared with those of Li$_2$ZrCl$_6$ and nZrO$_2$−2Li$_2$ZrCl$_6$ in Fig. 1c, d and Supplementary Fig. 5. Zr K-edge X-ray absorption near-edge structure (XANES) spectra for all three samples showed the main edge position at ~18020 eV (Supplementary Fig. 5a), confirming the tetravalent oxidation state of Zr[26]. A Zr K-edge extended X-ray absorption fine structure (EXAFS) spectrum of ZrO$_2$−2Li$_2$ZrCl$_6$ exhibited a distinct peak at ~1.5 Å (Fig. 1c) corresponding to the Zr-O coordination in ZrO$_2$[51], proving the

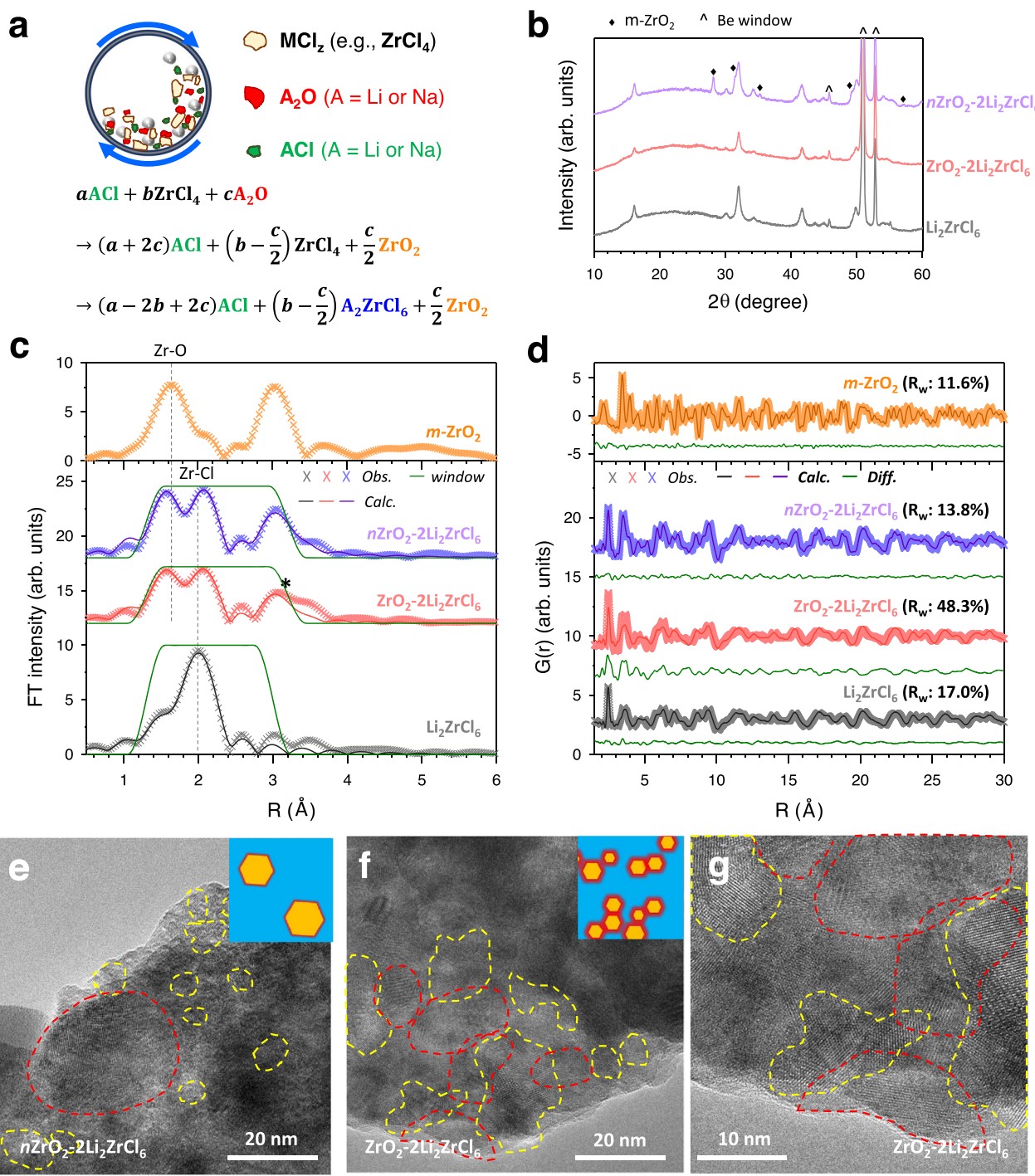

**Fig. 1 | Synthesis and characterization of HNSEs (ZrO₂−2Li₂ZrCl₆). a** Schematic of the one-pot mechanochemical synthesis of ZrO₂(·ACl)-A₂ZrCl₆ (A = Li or Na) HNSEs. **b**–**d** XRD patterns (**b**), coordination-number-refined Zr K-edge EXAFS fitting curves (**c**), and PDF G(r) with best-fit (**d**) for Li₂ZrCl₆, ZrO₂-2Li₂ZrCl₆, and nZrO₂-2Li₂ZrCl₆. **e**–**g** Cryo-HRTEM images of nZrO₂-2Li₂ZrCl₆ (**e**) and ZrO₂-2Li₂ZrCl₆ at lower magnification (**f**) and higher magnification (**g**). The red and yellow outlines indicate the domains of ZrO₂ and Li₂ZrCl₆, respectively. Schematics of the nanostructures (blue: Li₂ZrCl₆, yellow: ZrO₂, red: interface or interphase) are shown in the insets of (**e**) and (**f**).

mechanochemically driven formation of nanosized ZrO₂ via the reaction of ZrCl₄ with Li₂O. The EXAFS spectra of Li₂ZrCl₆, ZrO₂−2Li₂ZrCl₆, and nZrO₂−2Li₂ZrCl₆ also showed peaks at ~2 Å for the shortest Zr-Cl coordination, confirming the octahedral coordination of Zr (ZrCl₆²⁻)[26]. However, the local structural environments of Zr-O and Zr-Cl in the ZrO₂−2Li₂ZrCl₆ HNSE differ from those in single-phase ZrO₂ and Li₂ZrCl₆. While the Zr-O bond length decreases, the average Zr-Cl bond length slightly increases (from 2.46 Å to 2.47 Å in Supplementary Tables 2, 3) in ZrO₂−2Li₂ZrCl₆ HNSE compared with each of the ZrO₂ and Li₂ZrCl₆ phases. This result implies that the two nano-scale phases mutually affect the local structure of each other at the interface region. Such shrinkage of the ZrO₂ polyhedron and expansion of the Cl-containing polyhedron could enlarge the size of the Li⁺ transport channels in the ZrO₂−2Li₂ZrCl₆ HNSE, as discussed below for the

analysis of the DFT calculations. Furthermore, compared to $nZrO_2-2Li_2ZrCl_6$, the $ZrO_2-2Li_2ZrCl_6$ HNSE showed a broader peak at ~3 Å for Zr-Zr coordination (indicated by an asterisk in Fig. 1c). A twofold increase in the Debye-Waller factor (Supplementary Tables 2, 3) for such Zr-Zr bonding ($4.3 \times 10^{-3}$ Å$^2$ for $nZrO_2-2Li_2ZrCl_6$, $8.0 \times 10^{-3}$ Å$^2$ for $ZrO_2-2Li_2ZrCl_6$ HNSE) also suggested the highly disordered nature of the HNSE owing to the formation of a higher number of interfaces than in $nZrO_2-2Li_2ZrCl_6$. The refined coordination numbers (Supplementary Tables 2, 3) for Zr-O and Zr-Cl bonds in the HNSE were significantly less than the ideal values based on the model $ZrO_2$ and $Li_2ZrCl_6$ structures (e.g., 7.0 to 4.5 in Zr-O and 6.0 to 5.5 in Zr-Cl), providing additional evidence for the formation of interphases. The Cl K-edge XANES spectra (Supplementary Fig. 5b) were almost identical for all three samples with no pre-edge signals, indicating the ionic characteristics of the Zr-Cl bond[26].

The PDF G(r) values of $Li_2ZrCl_6$, $ZrO_2-2Li_2ZrCl_6$, and $nZrO_2-2Li_2ZrCl_6$ are shown in Fig. 1d. The refinement result for $Li_2ZrCl_6$ confirmed the hcp trigonal structure with the $P\bar{3}m1$ space group[26,33]. PDF G(r) for both $ZrO_2-2Li_2ZrCl_6$ and $nZrO_2-2Li_2ZrCl_6$ revealed signals for both trigonal $Li_2ZrCl_6$ and monoclinic $ZrO_2$. Importantly, whereas the $ZrO_2$ PDF signal for $nZrO_2-2Li_2ZrCl_6$ was distinct up to ~10 Å, that for the $ZrO_2-2Li_2ZrCl_6$ HNSE disappeared above ~5 Å. This result indicated that the mechanochemically derived $ZrO_2$ of the HNSE exhibited much smaller grain sizes and/or higher disorder compared with $nZrO_2-2Li_2ZrCl_6$[33]. Moreover, we performed PDF fitting analysis for $ZrO_2-2Li_2ZrCl_6$ HNSE and $nZrO_2-2Li_2ZrCl_6$ using the model crystal structures obtained by fits of the single-phase PDFs of $Li_2ZrCl_6$ and $ZrO_2$. The fit of $nZrO_2-2Li_2ZrCl_6$ converged and showed a reasonably decent fit ($R_w = 13.8$ %) using two model structures. However, the fit of $ZrO_2-2Li_2ZrCl_6$ HNSE was unsuccessful, ending with an unacceptable $R_w$ value of 48.3%, which suggested the necessity of including an additional (inter)phase to unambiguously describe its complicated structure. This result indirectly emphasized the formation of interphases, enough to affect the average structure of the $ZrO_2-2Li_2ZrCl_6$ HNSE. Further refinements of the PDF using additional (inter)phases indicated by DFT calculations are discussed in the Interfacial Superionic Conduction of HNSEs section.

Cryogenic high-resolution transmission electron microscopy (cryo-HRTEM) images for $nZrO_2-2Li_2ZrCl_6$ and $ZrO_2-2Li_2ZrCl_6$ are displayed in Fig. 1e-g. The corresponding fast Fourier transform (FFT) images are provided in Supplementary Figs. 6 and 7. For $nZrO_2-2Li_2ZrCl_6$, crystalline $ZrO_2$ nanoparticles with sizes 20-50 nm were distributed in the glass-ceramic $Li_2ZrCl_6$ matrix (Fig. 1e). Interestingly, the cryo-HRTEM images of the $ZrO_2-2Li_2ZrCl_6$ HNSE (Fig. 1f, g) showed that $ZrO_2$ nanograins with <20 nm sizes formed a local percolating network nanostructure. $Li_2ZrCl_6$ nanograin domains <10 nm were embedded in the percolating network nanostructure, implying large-area percolating interfaces (Supplementary Note 3), which were critical for anomalous interfacial conduction[46,47]. The mechanochemical preparation of the HNSEs was also effective for the Na$^+$ analogues, such as $ZrO_2-2Na_2ZrCl_6$ and $0.13ZrO_2-0.61NaCl-0.26Na_2ZrCl_6$. The corresponding XRD, PDF, and HRTEM results are presented in Supplementary Figs. 8 and 9.

The ionic conductivity results for the Li$^+$ and Na$^+$ HNSEs obtained by the AC impedance method using ion-blocking Ti|SE|Ti symmetric cells are shown in Fig. 2. Nyquist and Arrhenius plots for the ionic conductivity are displayed in Fig. 2a-c. The equivalent circuit model and the fitted results are also provided in Supplementary Fig. 10a and Supplementary Table 4. Compared to $Li_2ZrCl_6$, the $ZrO_2-2Li_2ZrCl_6$ HNSE exhibited approximately threefold enhancement in the Li$^+$ conductivity (1.1 vs. 0.40 mS cm$^{-1}$) with a lowered activation energy (0.31 vs. 0.37 eV), which was noteworthy in that the ionically insulating $ZrO_2$ occupied theoretically as much as 7.86 vol.% in $ZrO_2-2Li_2ZrCl_6$, based on the chemical formula. Interestingly, the Li$^+$ conductivity of the control sample, $nZrO_2-2Li_2ZrCl_6$ (0.6 mS cm$^{-1}$), was also slightly higher

than that of $Li_2ZrCl_6$ but lower than that of the HNSE $ZrO_2-2Li_2ZrCl_6$. Considering the smaller sizes of $ZrO_2$ grains and thus, larger interfacial areas for $ZrO_2-2Li_2ZrCl_6$, this result implies increased Li$^+$ conduction at the $ZrO_2-2Li_2ZrCl_6$ interfaces[42]. Although the Li$^+$ conductivities for nanocomposites of $Li_2ZrCl_6$ with different metal oxide nanoparticles were consistently higher than that of $Li_2ZrCl_6$, none of them were as high as the conductivity of the mechanochemically prepared $ZrO_2-2Li_2ZrCl_6$ HNSE (Supplementary Fig. 11). To the best of our knowledge, the Li$^+$ HNSE was the first inorganic superionic conductor that exploited the interfacial effect to promote conduction with ionic conductivity reaching 1 mS cm$^{-1}$ at 25-30 °C. Furthermore, the maximum ionic conductivity of the HNSE Na$^+$ analogues was approximately an order of magnitude greater (0.11 mS cm$^{-1}$ for $0.13ZrO_2-0.61NaCl-0.26Na_2ZrCl_6$) than that of $Na_2ZrCl_6$ (0.011 mS cm$^{-1}$) (Fig. 2b, c, and Supplementary Fig. 12). This value is the highest among the Na$^+$ halide SEs developed thus far[32-34].

The extended compositions of Li$^+$ and Na$^+$ HNSEs were further explored, as shown in the contour map of ionic conductivity for the ternary system of $ZrO_2$-ACl-$A_2ZrCl_6$ (A = Li or Na) with the volume fraction scale (Fig. 2d). The corresponding Li$^+$ and Na$^+$ conductivities are summarized in Supplementary Tables 5 and 6. Four features are worth noting. First, the ionic conductivities of the HNSEs were enhanced as the $ZrO_2$ fraction was increased. Second, compared to $Li_2ZrCl_6$ (or $Na_2ZrCl_6$), $0.53LiCl-Li_2ZrCl_6$ (or $0.53NaCl-Na_2ZrCl_6$) showed higher ionic conductivities of 0.70 vs. 0.40 mS cm$^{-1}$ for Li$^+$, implying enhanced ionic conduction at the ACl-$A_2ZrCl_6$ interfaces. However, from the result where the LiCl/$ZrO_2$ ratio varied with the fixed $Li_2ZrCl_6$ fraction, $ZrO_2$ was more effective than LiCl in enhancing the ionic conductivity of the HNSEs. Third, the maximum ionic conductivities of the HNSEs were found for the ternary HNSEs of $0.44ZrO_2-1.26LiCl-0.56Li_2ZrCl_6$ (1.3 mS cm$^{-1}$) and $0.13ZrO_2-0.61NaCl-0.26Na_2ZrCl_6$ (0.11 mS cm$^{-1}$). Finally, an accessible compositional area was restricted to Region I (Fig. 2d). Alternative oxygen sources can be used to access the region beyond the upper boundary limit of Region I, leading to a further enhancement of the ionic conductivities.

## Interfacial superionic conduction of HNSEs

In many previous reports about AX/metal oxide systems (A = Li, Cu, Ag; X = Cl, Br, I) with behaviour similar to that of the HNSEs, anomalously high ionic conductivities were attributed to the space charge layer (SCL) effect, which has been an issue of debate[42,47]. The ionic conduction between Li$^+$ conductors and metal oxides (non-Li$^+$-conducting materials) can be improved by more charge carriers at the interfaces through SCL[42]. However, the superionic conduction of HNSE could not be explained solely by the conventional SCL effect observed in AX/metal oxide systems due to their much lower ionic conductivities (10$^{-1}$ to 10$^{-3}$ mS cm$^{-1}$ at 25 °C). To elucidate the underlying mechanism for the enhanced $ZrO_2$/$Li_2ZrCl_6$ interfacial superionic conduction of HNSEs, we conducted DFT calculations.

We considered a small amount ($x = 0.5$ in $Li_{2+x}ZrCl_{6-x}O_x$) of anion exchange between $Li_2ZrCl_6$ (LZC) and $ZrO_2$ at the interface during synthesis and an excess Li concentration at the LZC side for local charge compensation with oxygen substitution (see Supplementary Fig. 13 and Supplementary Note 4). Figure 3a and Supplementary Fig. 14 show our model structures of LZC, $Li_{2.5}ZrCl_{5.5}O_{0.5}$ (LZCO), $ZrO_2$, and $ZrO_{2-x}Cl_x$. After structural relaxation using DFT calculations, the structure of LZCO showed elongation of the average bond length of Zr-Cl compared with that of LZC, and Cl-substitution of $ZrO_2$ drove an increase in the Zr-Zr distance (Supplementary Table 7 and Supplementary Fig. 15). These structural changes with anion exchange between LZC and $ZrO_2$ agreed with the experimental observation in Zr K-edge EXAFS results (Fig. 1c, Supplementary Table 2), which demonstrated a slight increase in Zr-Cl bond length (from 2.46 to 2.47 Å) and broadening of the Zr-Zr peak (~3.5 Å) for the HNSE. The distances among $ZrCl_{6-x}O_x$ octahedra in LZCO increased due to larger

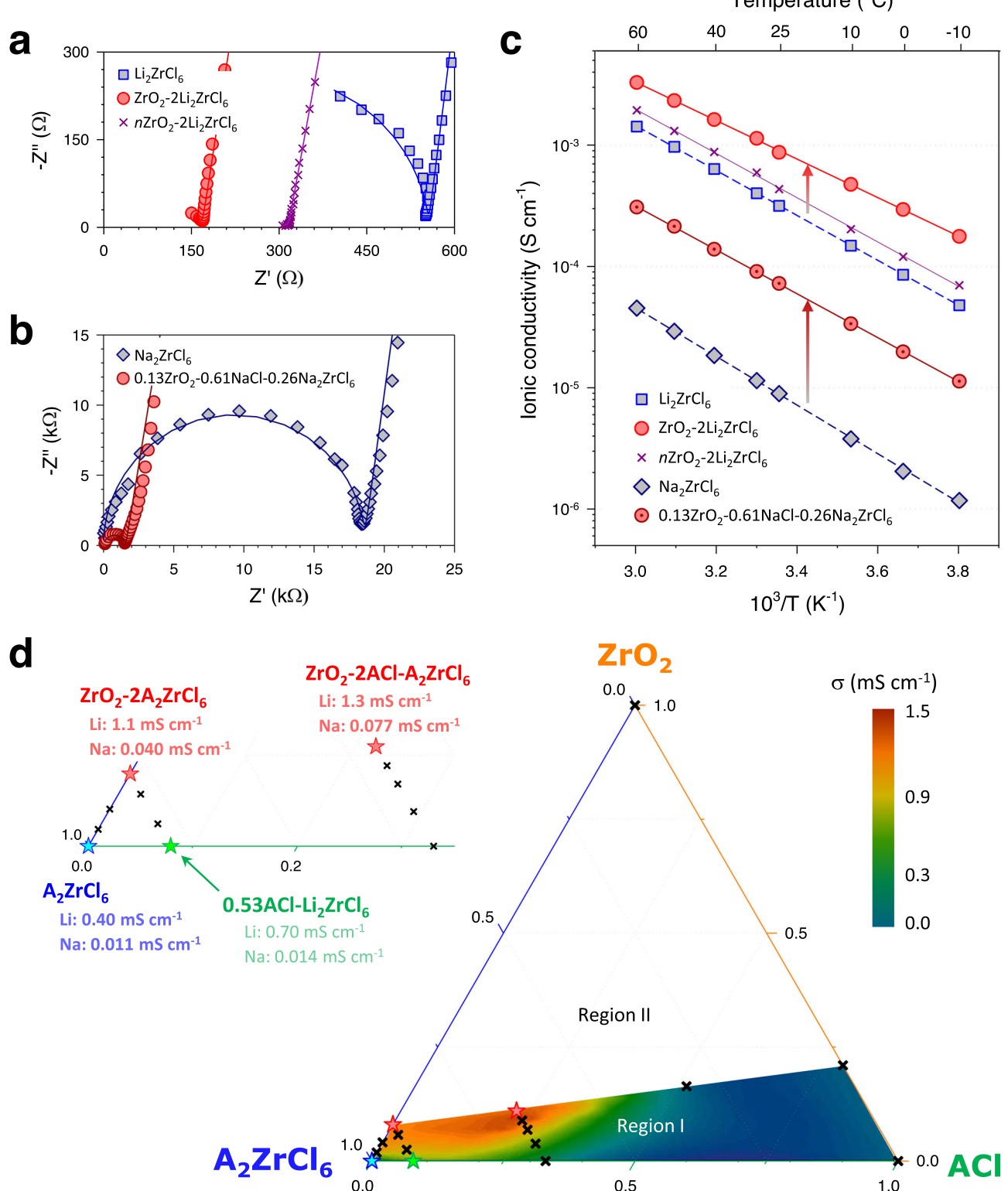

**Fig. 2 | Ionic conductivities of Li+ and Na+ HNSEs (ZrO₂(-ACl)-A₂ZrCl₆).**
**a**, **b** Nyquist plots of ion-blocking Ti|SE|Ti symmetric cells at 30 °C for Li+ HNSEs (ZrO₂-2Li₂ZrCl₆ vs. Li₂ZrCl₆, nZrO₂-2Li₂ZrCl₆) (**a**) and Na+ HNSE (0.13ZrO₂-0.61NaCl-0.26Na₂ZrCl₆ vs. Na₂ZrCl₆) (**b**). The fitted lines using equivalent circuit model in Supplementary Fig. 10 with raw data (symbol) are also shown in (**a**) and (**b**). **c** Arrhenius plots of ionic conductivities for conventional halide SEs and HNSEs. **d** Ionic conductivity contour plot at 30 °C for ternary ZrO₂-ACl-A₂ZrCl₆ HNSEs. The contour map was plotted using data represented by stars and crosses, with further details provided in Supplementary Tables 5 and 6.

electrostatic repulsion of Cl⁻–O²⁻ than that of Cl⁻–Cl⁻, resulting in lattice volume expansion (Supplementary Fig. 16). A Zr-O bond that was shorter than the Zr-Cl bond drove the shrinkage of ZrCl₆₋ₓOₓ octahedra (Supplementary Table 7). These two factors together

enlarged the Li+ transport channel (Fig. 3b), which boosted the migration of Li ions. In addition, the Li+ concentration was higher in LZCO than in LZC, which enriched the ionic carrier concentration. Notably, LZCO was energetically less favourable than the composite of

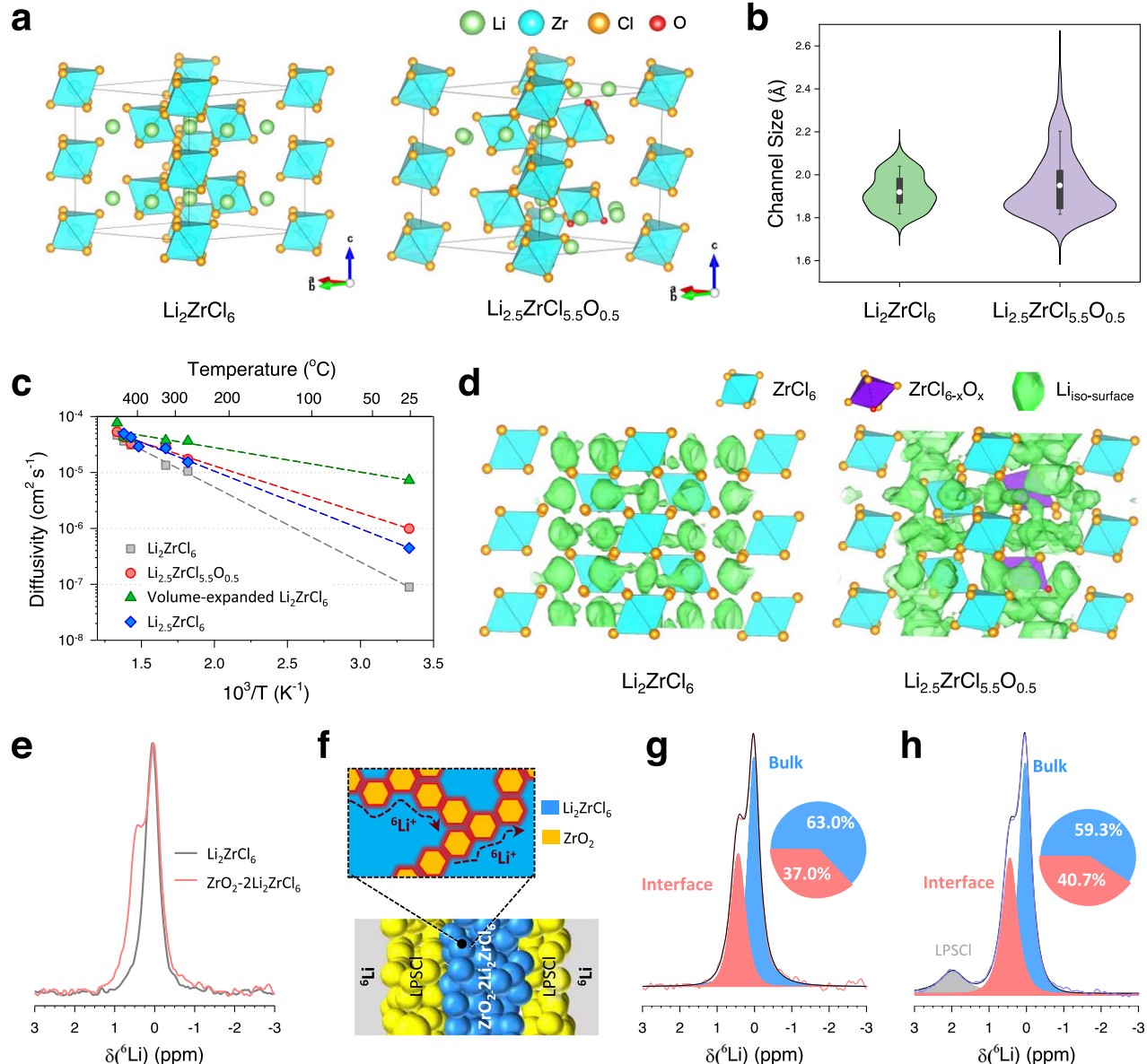

**Fig. 3 | Interfacial Li⁺ superionic conduction in HNSEs. a** Crystal structures of $Li_2ZrCl_6$ and $Li_{2.5}ZrCl_{5.5}O_{0.5}$. **b** Topological analysis and channel size of $Li_2ZrCl_6$ and $Li_{2.5}ZrCl_{5.5}O_{0.5}$. The white circle, box and whisker indicate the mean value, values from the 25% to the 75% percentiles and values from the 10% to the 90% percentiles, respectively. **c** Arrhenius plots of the AIMD simulation extrapolated to 300 K. **d** Li probability density at 600 K in ~200 ps (isosurface value $P = P_{max}/50$). **e** ⁶Li MAS-NMR spectra for $Li_2ZrCl_6$ and $ZrO_2$-$2Li_2ZrCl_6$. **f** Schematic of ⁶Li|LPSCl|($ZrO_2$-$2Li_2ZrCl_6$)|LPSCl|⁶Li symmetric cells with interfacial superionic Li⁺ diffusion pathways. (Blue: $Li_2ZrCl_6$, yellow: $ZrO_2$, red: interface or interphase) **g, h** ⁶Li MAS-NMR spectra of $ZrO_2$-$2Li_2ZrCl_6$ before (**g**) and after cycling (**h**).

$ZrCl_4$, LiCl, and $ZrO_2$ (Supplementary Table 8), indicating that it would be challenging to synthesize LZCO directly in a general reaction. However, its energy above the hull is not that high; thus, it can be partially formed at the interface by high-energy mechanochemical synthesis.

To further verify the composition of $ZrO_2$−$2Li_2ZrCl_6$ and validate the O-substituted LZCO populated at the interfaces, we performed PDF fitting across different refinement ranges (low r range of 1.5–10 Å; high r range of 10−30 Å) as shown in Supplementary Fig. 17 and Supplementary Table 9. After extensive preliminary fitting attempts using various combinations of model structures, including the LZCO interphase structure provided via DFT calculations (Supplementary Table 7), the best-fit result ($R_w = 10.9\%$) is achieved by including the LZCO interphase with minor $Li_2O$ impurity (Supplementary Table 10) in the low r range (1.5 ~ 10 Å), where the interface regime

becomes prevailing. On the other hand, the medium and average structure in the high r range (10 ~ 30 Å) can be well represented by a single $Li_2ZrCl_6$ structure. The calculated composition obtained by the PDF fit corresponds to $1.47Li_2ZrCl_2$−$0.36Li_{2.5}ZrCl_{5.5}O_{0.5}$(interphase) $−1.01ZrO_2$−$0.16Li_2O$ (Supplementary Table 11). This result strongly supports the presence of a LZCO interphase between $ZrO_2$ and LZC nanodomains.

Ab initio molecular dynamics (AIMD) simulations were also performed to verify superionic conduction at the HNSE interface (Fig. 3c, Supplementary Fig. 18 and Table 12). LZCO shows a faster Li⁺ diffusion than LZC at all temperatures for which AIMD simulations were performed, but also has a gentle slope compared with the Li⁺ diffusivities of LZC. It is expected that Li⁺ diffusion shows ~11 times faster for LZCO than LZC at 300 K. To explain the origin of such fast diffusion, we generated two hypothetical structures of LZC: (1) off-stoichiometric

Li-rich LZC ($Li_{2.5}ZrCl_6$), where the amount of Li in the structure was simply increased, and (2) volume-expanded LZC, where the lattice parameters were the same as those of LZCO. The diffusivities of both hypothetical structures at 300 K were higher than that of the LZC structure (Fig. 3c), implying that both volume expansion and Li enrichment positively impacted facile diffusion. The Li probability density showed a more 3D-connected and broadened Li isosurface in LZCO than the LZC case, revealing that the overall expanded $Li^+$ diffusion pathway activated the movement of $Li^+$ (Fig. 3d and Supplementary Fig. 19). In addition, we discuss more detailed observations of the $Li^+$ diffusion behaviour near the neighbouring anions ($O^{2-}$, $Cl^-$) of LZCO in Supplementary Fig. 20 and the Supplementary Note 5. To effectively exploit interfacial superionic conduction, it is crucial to not only accumulate $Li^+$ carriers at the interface but also widen $Li^+$ channels via anion substitution. Notably, the interface formed by a poor ionic conductor (LiCl) and superionic conductor ($Li_2ZrCl_6$) only promoted Li accumulation by the chemical potential difference, excluding the effect of the channel size increase. This may result in an insignificant increase in ionic conduction compared to that in the HNSE of $ZrO_2$-LZC, as shown in our experimental measurements (Fig. 2d). More $ZrO_2$-LZC interfaces in HNSEs result in more interfacial phases (LZCO), further boosting the superionic conduction.

To probe the local $Li^+$ environments at the interfaces of the HNSEs, $^6Li$ MAS-NMR measurements were conducted for $Li_2ZrCl_6$ and $ZrO_2-2Li_2ZrCl_6$. For both spectra, the main signals are shown at 0.05 ppm corresponding to $Li_2ZrCl_6$ (Fig. 3e). However, in sharp contrast to $Li_2ZrCl_6$, the $ZrO_2-2Li_2ZrCl_6$ HNSE spectrum exhibited a distinct peak at -0.42 ppm. In a previous study on $LiBH_4/Al_2O_3$, a similar shoulder peak was observed and attributed to the highly conductive interface region affected by $Al_2O_3$[48]. Because the electronegativity of oxygen is larger than that of chlorine, electrons in $Li^+$ at the interfaces will be withdrawn more for HNSE than for $Li_2ZrCl_6$, which implies more deshielding and thus explains the evolution of the peak at the higher chemical shift[48,52]. In this regard, the shoulder peak at -0.42 ppm for the HNSE likely corresponded to the O-substituted interphase suggested by the DFT calculations. Furthermore, the $Li^+$ migration pathways in the nanocomposite structure of $ZrO_2-2Li_2ZrCl_6$ were investigated by $^6Li$ exchange experiments using $^6Li|LPSCl|HNSE|LPSCl|$ $^6Li$ symmetric cell (Fig. 3f)[53]. After repeated cycling (Supplementary Fig. 21), the $^6Li$ NMR spectrum of the HNSE was compared with the result for pristine HNSE (Fig. 3g, h). The area fraction of the interphase peak increased from 37.0 to 40.7% after cycling, which corroborated the promoted interfacial $Li^+$ conduction.

## General applicability of the HNSE strategy in interfacial conduction

The material space for the HNSEs is expandable beyond the $ZrO_2$-$ACl$-$A_2ZrCl_6$ system, as illustrated schematically in Supplementary Fig. 22. Two-step mechanochemical protocols can be used to produce multimetal HNSEs. In the first step, $Li_2O$ and metal chloride ($MCl_y$) react to form $MO_x$-LiCl nanocomposites. The subsequent step involves the reaction with additional metal halides ($M'X_y$ (with $M''X_y$)) to form multimetal HNSEs, such as $MO_x$-LiCl-$Li_aM'Cl_b$ and $MO_x$-LiCl-$Li_aM'M''Cl_b$. Following this two-step path, $Al_2O_3-3Li_2ZrCl_6$ and $SnO_2-2Li_2ZrCl_6$ HNSEs were prepared (Supplementary Fig. 23). Similar to $ZrO_2-2Li_2ZrCl_6$, the $Al_2O_3-3Li_2ZrCl_6$ and $SnO_2-2Li_2ZrCl_6$ HNSEs exhibited enhanced $Li^+$ conductivities of 0.88 and 1.6 mS $cm^{-1}$, respectively, compared to that of $Li_2ZrCl_6$ (Supplementary Fig. 23). Notably, the single-step protocol using a mixture of $Li_2O$, $AlCl_3$, and $ZrCl_4$ resulted in a poor $Li^+$ conductivity of 0.3 mS $cm^{-1}$ due to the formation of an unfavourable LiCl component. In our previous study, the $Li^+$ conductivity of $Li_2ZrCl_6$ was improved via aliovalent substitution with $Fe^{3+}$, showing a maximum conductivity of 1.0 mS $cm^{-1}$ ($Li_{2.25}Zr_{0.75}Fe_{0.25}Cl_6$)[26]. Such HNSEs can also be prepared by the twostep protocol. To avoid the reaction of $FeCl_3$ with $Li_2O$, after

$ZrO_2-2Li_2ZrCl_6$ was prepared, $FeCl_3$ and LiCl were added to substitute $Fe^{3+}$ in $Li_2ZrCl_6$. $Li_{2+x}Zr_{1-x}Fe_xCl_6$ showed enhanced $Li^+$ conductivities for HNSEs over the entire range of $x$, reading a maximum $Li^+$ conductivity of 1.4 mS $cm^{-1}$ (Supplementary Fig. 24).

Importantly, owing to the small ionic size and low polarizability of $F^-$ ($r(F^-) = 119$ pm, $r(Cl^-) = 181$ pm, ionic radius values represent the crystal ionic radii[54]), F-substitution in SEs generally decreases the ionic conductivity[37,38], but it can be counterbalanced by applying the HNSE synthetic strategy. From the XRD results, the F-substitution limit in $Li_2ZrCl_{6-x}F_x$ is approximately $x = -1.0$ (Supplementary Fig. 25). The XRD pattern of the $ZrO_2-2Li_2ZrCl_5F$ HNSE showed a slight positive shift of the (301) peak at -32° (Fig. 4a), indicating the successful fluorination of the $Li_2ZrCl_6$ domain. Consistent with the result for the $ZrO_2-2Li_2ZrCl_6$ HNSE, the $ZrO_2-2Li_2ZrCl_5F$ HNSE also exhibited a shoulder peak in $^6Li$ MAS-NMR spectrum (Supplementary Fig. 26). This result suggests an O-substituted $Li_2ZrCl_5F$ at the interface as revealed for $ZrO_2-2Li_2ZrCl_6$. The HRTEM results are also provided in Supplementary Fig. 27. While the $Li^+$ conductivity of $Li_2ZrCl_6$ was decreased to 0.35 mS $cm^{-1}$ by fluorination ($Li_2ZrCl_5F$), the $ZrO_2-2Li_2ZrCl_5F$ HNSE exhibited even higher $Li^+$ conductivity of 0.49 mS $cm^{-1}$ compared to $Li_2ZrCl_6$ (0.40 mS $cm^{-1}$, Fig. 4b). Notably, when assessed by cyclic voltammetry (CV) tests at 30 °C, $ZrO_2-2Li_2ZrCl_5F$ exhibited a smaller integrated current of 1.98 mA V $g^{-1}$ up to 5.0 V (vs. $Li/Li^+$, although all cells in this work utilized a Li-In alloy electrode instead of metallic Li, we report the voltage vs. $Li/Li^+$ instead of vs. $Li$-$In/Li^+$ as a convention in the electrochemistry field. Hereafter, it is to be understood that all reported cell voltages vs. $Li/Li^+$ as being shifted by 0.62 V from the cell voltage vs. $Li$-$In/Li^+$ for better comparison with literature data. Further detailed discussion is provided in Supplementary Note 6, Supplementary Fig. 28), compared to $Li_2ZrCl_6$ (2.76 mA V $g^{-1}$, Fig. 4c, Supplementary Table 13). The difference became even larger at the second cycle (0.55 vs. 2.00 mA V $g^{-1}$ for $ZrO_2-2Li_2ZrCl_5F$ and $Li_2ZrCl_6$, respectively). Furthermore, $Li_2ZrCl_6$ showed a cathodic peak at ≈3.5 V (vs. $Li/Li^+$) at the first cycle and they intensified further at the second cycle, which is indicated by an asterisk in Fig. 4c. It is speculated that the cathodic currents originate from the decomposition byproducts formed during the prior positive scan[55]. By contrast, $ZrO_2-2Li_2ZrCl_5F$ exhibited negligible cathodic currents, which agrees with the substantially lower oxidation currents compared with $Li_2ZrCl_6$, thus suggesting the passivating behaviour of $ZrO_2-2Li_2ZrCl_5F$. The DFT results consistently revealed that the oxidative limit of $Li_2ZrCl_5F$ (4.274 V vs. $Li/Li^+$) was slightly lower than that of $Li_2ZrCl_6$ (4.307 V vs. $Li/Li^+$), but the formation of desirable F-based passivating interphase materials such as $Li_2ZrF_6$ and $Li_3Zr_4F_{19}$ can increase the range of the anodic limit (Fig. 4d and Supplementary Table 14)[37,38].

## Electrochemical energy storage performances of all-solid-state Li-based cells with HNSEs

The mechanochemically prepared HNSEs $ZrO_2$-$2Li_2ZrCl_6$ and $ZrO_2$-$2Li_2ZrCl_5F$ (hereafter referred to as $ZrO_2$-LZC and $ZrO_2$-LZCF, respectively) were tested in combination with uncoated LCO and S-NCM88 positive electrodes in ASSB cells, and the results were compared to those obtained using $Li_2ZrCl_6$ (hereafter referred to as LZC) as shown in Fig. 5 and Supplementary Fig. 29. LPSCl monolayers or ($ZrO_2$-LZCF)| LPSCl bilayers and Li-In counter electrodes were employed for the ASSB cells (Fig. 5a). At 30 °C and a cut-off voltage of 4.3 V (vs. $Li/Li^+$), all three LCO cells exhibited good performances with marginal differences in terms of their capacity, initial Coulombic efficiency (ICE), and capacity retention at 82.0 mA $g^{-1}$ (Supplementary Fig. 30, Supplementary Table 15). However, upon increasing the temperature to 60 °C, the performance difference became distinct in the descending order of $ZrO_2$-LZCF > $ZrO_2$-LZC » LZC (Fig. 5b, c, Supplementary Fig. 29a); the ICEs were 94.5%, 91.8%, and 80.3%, and the values of the capacity retention at the 100th cycle were 93.7%, 68.0%, and 1.7%, respectively. When other major halide SEs like $Li_3YCl_6$ and $Li_3InCl_6$

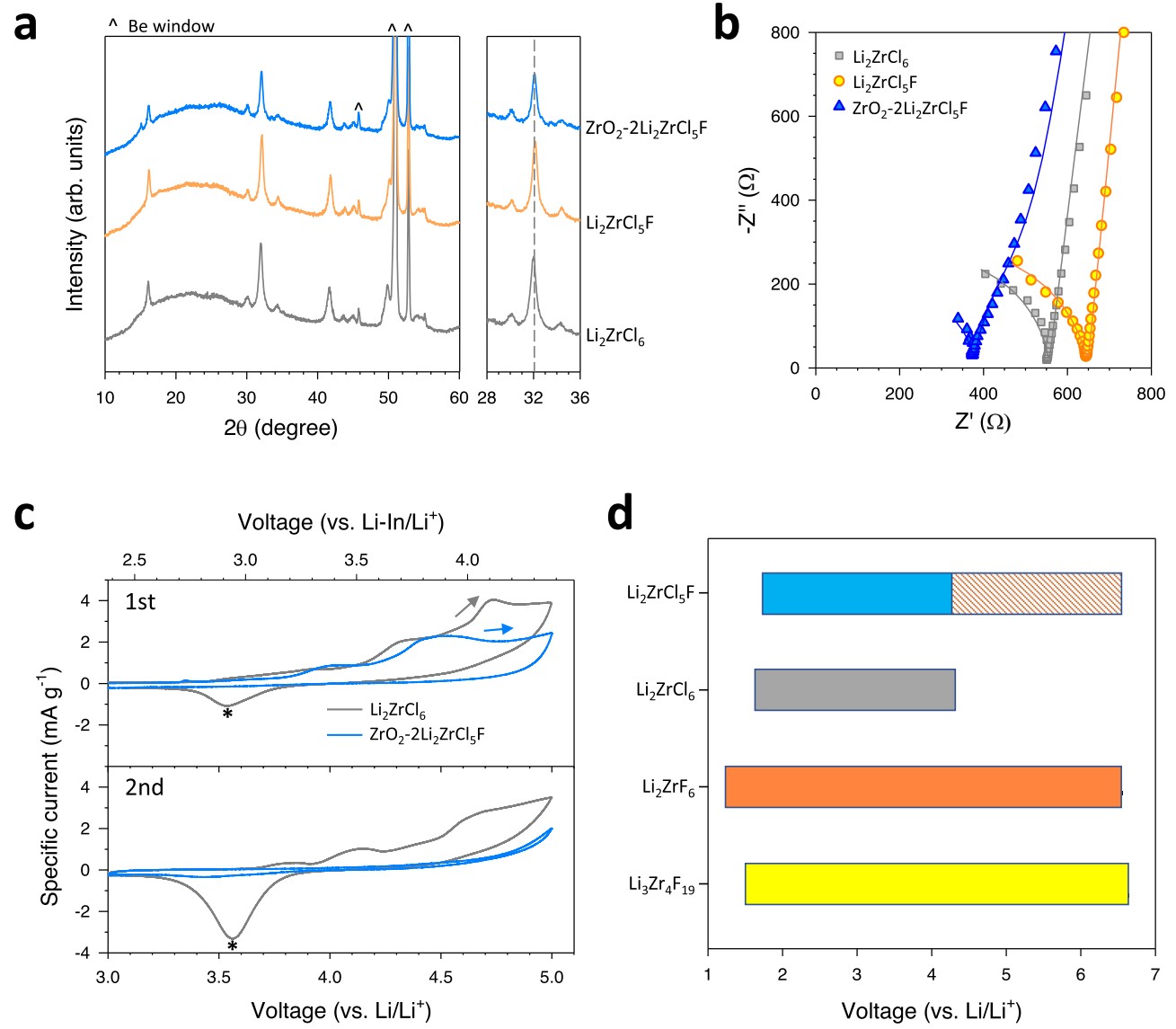

**Fig. 4 | HNSE for F-substituted Li₂ZrCl₆ (ZrO₂-2Li₂ZrCl₅F). a–c** XRD patterns (**a**) and Nyquist plots of Ti|SE|Ti symmetric cells (**b**) for $Li_2ZrCl_6$, $Li_2ZrCl_5F$, and $ZrO_2$-$2Li_2ZrCl_5F$. The fitted lines are also plotted in (**b**). The symbols represent the raw data, and the lines correspond to the fitted results obtained from the equivalent circuit model, as described in Supplementary Fig. 10. **c** CV curves for (SE-C)|SE| LPSCl|(Li-In) cells between 3.0 and 5.0 V (vs. Li/Li⁺) at 0.1 mV s⁻¹ and 30 °C.

**d** Calculated electrochemical stability of $Li_2ZrCl_6$, $Li_2ZrCl_5F$, $Li_2ZrF_6$ and $Li_3Zr_4F_{19}$. $Li_2ZrF_6$ and $Li_3Zr_4F_{19}$ are thermodynamically stable passivating interphases that can be decomposed from $Li_2ZrCl_5F$. The dashed box of $Li_2ZrCl_5F$ represents the electrochemical stability window stabilized by the decomposition products of $Li_2ZrCl_5F$ ($Li_2ZrF_6$ and $Li_3Zr_4F_{19}$). Calculation details are found in the "Methods" section.

were tested at 60 °C, capacity fading was also observed (Supplementary Fig. 31). The poor cycling performances of LCO electrodes using halide SEs were uncommon[7,20,25,26,31,38]. Despite being overlooked, these results could be associated with compatibility with the sulfide SE used for the SE layer[41]. The incompatibility between sulfide and halide SEs impacted Coulombic efficiency. For monolayer cells utilizing LZC or ZrO₂-LZC at 30 °C and 60 °C (Supplementary Figs. 29 and 30), Coulombic efficiency values continuously increased beyond 100%, suggesting side reactions between the sulfide and halide SEs. The good capacity retention and consistent Coulombic efficiency with ZrO₂-LZCF indicated the compatibility of ZrO₂-LZCF with LPSCl. The cells using the LZC catholyte with the (ZrO₂-LZCF)|LPSCl bilayer, where direct contact between LZC and LPSCl was prevented (Fig. 5d, e, Supplementary Fig. 29b), consistently showed degrading but improved overall performance, from 1.7% to 70.0% for the capacity retention at the 100th cycle. In contrast, for the cells with the LCO electrodes employing fluorinated HNSE ZrO₂-LZCF, the cycling

performance was satisfactory, regardless of the separating SE layer, with capacity retention of 93.7% and 93.4% after 100 cycles using mono- and bilayers, respectively. The corresponding electrochemical impedance spectroscopy (EIS) results are provided in Supplementary Fig. 32 and the equivalent circuit and the fitted results with detail discussion are also provided in Supplementary Fig. 10b, Supplementary Table 16 and Supplementary Note 7. The still degrading performance of the cells using LZC with the (ZrO₂-LZCF)|LPSCl bilayer originates from its electrochemical instability and/or incompatibility with cathode material (LCO). Based on the results obtained thus far, we conclude that the reason for the poor performance of the LZC|(LPSCl monolayer) combination at 60 °C was the incompatibility of LZC with LPSCl and LiMO₂, and electrochemical instability, whereas ZrO₂-LZCF is compatible and electrochemically stable.

DFT calculations were conducted to assess the compatibilities of LCO and sulfide SEs with halide SEs. While the mixtures of LCO and halide SEs ($Li_2ZrCl_6$, $Li_3YCl_6$, $Li_3InCl_6$) showed low mutual reaction

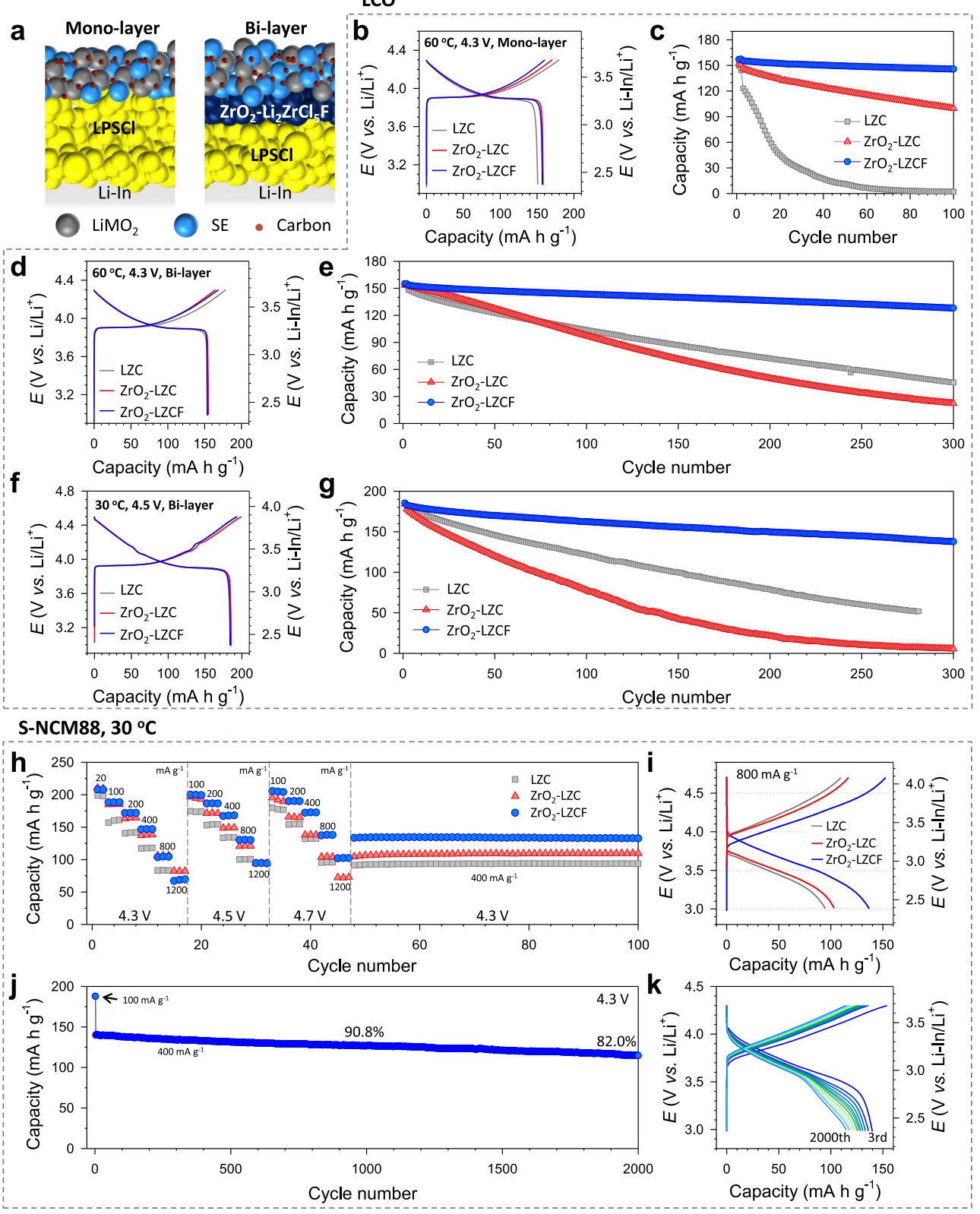

**Fig. 5 | LCO and S-NCM88 ASSB cells employing HNSEs (ZrO₂-2Li₂ZrCl₆ and ZrO₂-2Li₂ZrCl₅F vs. Li₂ZrCl₆).** **a** Schematics of ASSB cells employing an LPSCl monolayer or (ZrO₂-LZCF)|LPSCl bilayer. **b, c** First-cycle charge–discharge voltage profiles at 16.4 mA g⁻¹ and 60 °C (**b**) and cycling performances at 82.0 mA g⁻¹ (**c**) for LCO electrodes with an LPSCl monolayer cycled up to 4.3 V (vs. Li/Li⁺). **d–g** First-cycle charge–discharge voltage profiles at 16.4 mA g⁻¹ and cycling performances at 82.0 mA g⁻¹ for LCO electrodes with a ZrO₂-LZCF bilayer cycled up to 4.3 V at 60 °C (**d, e**) and 4.5 V (vs. Li/Li⁺) at 30 °C (**f, g**). **h, i** Rate capabilities with varying cut-off voltages (**h**) and the corresponding charge–discharge voltage profiles at 800 mA g⁻¹

(**i**) for S-NCM88 electrodes. **j, k** Cycling performance for the S-NCM88 electrode using ZrO₂-LZCF at 400 mA g⁻¹ (**j**) and the corresponding charge–discharge voltage profiles (**k**). LPSCl monolayers were used for the S-NCM88 cells. The capacity retention was determined by comparing capacities at the 1000th and 2000th cycles to that at the 3rd cycle. For the LCO-based electrodes, the specific capacity of 164 mA g⁻¹ corresponds to 1.30 mA cm⁻² and, for the S-NCM88-based electrodes the specific capacity of 200 mA g⁻¹ corresponds to 1.09 mA cm⁻². The specific current and capacity were determined based on the mass of active material (10.2 mg for LCO and 7.3 mg for S-NCM88). All the cells were cycled under a pressure of 70 MPa.

energies ($|\Delta E_{D, mutual}| < 0.4$ eV/formula unit (f.u.)), the halide/LPSCl SE mixtures exhibited high reaction energies over 0.4 eV/f.u., indicating their poor compatibility (Supplementary Table 17). For LZC and ZrO$_2$-LZCF, an EIS experiment using Ti|(halide-LPSCl mixture)|Ti cells stored at 60 °C showed marginal differences in Nyquist plots after a week (Supplementary Fig. 33), indicating a stable halide-LPSCl interface under no electrochemical driving forces (Supplementary Note 8). A control EIS experiment was also performed using Li-In|LPSCl|halide-carbon cells charged to 4.3 V (vs. Li/Li$^+$) at 60 °C so that the halide-LPSCl interfaces are subjected to the high voltage (Supplementary Fig. 34). From qualitative analysis of the Nyquist plots, it can be seen that the impedance continuously increased for the cells using LZC. In contrast, the impedance increased for an hour and then saturated for the cell comprised of ZrO$_2$-LZCF, which indicated good passivating behaviour. In summary, the halide/sulfide incompatibility, which has often been overlooked, was identified clearly, and it was found to be driven electrochemically at elevated temperatures. Importantly, F-substituted chloride SEs are free from this halide/sulfide incompatibility issue and thus allow for the use of sulfide monolayers, which simplifies the fabrication of ASSB cells for practical applications, as illustrated in Supplementary Fig. 35. High-voltage stabilities of up to 4.5 V (vs. Li/Li$^+$) for LCO electrodes using HNSEs were also tested in ASSB cells with bilayer at 30 °C (Fig. 5f, g and Supplementary Fig. 29c). The capacity retention with ZrO$_2$-LZCF was much higher (82.2% after 200 cycles) than that with LZC or ZrO$_2$-LZC (43.8% and 12.1%, respectively), which agreed well with the results of CV (Fig. 4c) and DFT calculations. The charge–discharge voltage profiles for the LCO electrodes at different cycles are shown in Supplementary Fig. 36.

The high-voltage stability of ZrO$_2$-LZCF enables pushing the upper cut-off voltage limit, offering an extra margin that counteracts polarization-driven capacity loss[56]. This feature could be advantageous for fast charging (Supplementary Fig. 37)[38]. Moreover, the elevated ion conduction in the HNSEs further boosts the fast-charging capability. The rate capabilities of cracking-free S-NCM88 positive electrodes in combination with Li-In negative electrodes were thus assessed using ASSB cells with LPSCl monolayers and stepwise increasing cut-off voltages of 4.3, 4.5, and 4.7 V (vs. Li/Li$^+$) at various specific currents and 30 °C (Fig. 5h, i, and Supplementary Fig. 38). The S-NCM electrodes showed high initial discharge capacities of 199, 210, and 208 mA h g$^{-1}$ at 20.0 mA g$^{-1}$ for LZC, ZrO$_2$-LZC, and ZrO$_2$-LZCF, respectively. At the lowest cut-off voltage of 4.3 V, the rate capabilities decreased in the following order: ZrO$_2$-LZC ≈ ZrO$_2$-LZCF > LZC, which was consistent with the Li$^+$ conductivity order (ZrO$_2$-LZC: 1.1 mS cm$^{-1}$ > ZrO$_2$-LZCF: 0.49 mS cm$^{-1}$ > LZC: 0.40 mS cm$^{-1}$) and the stability of LZCF. At higher cut-off voltages, especially 4.7 V, the electrodes employing ZrO$_2$-LZCF outperformed the others, emphasizing the effect of high-voltage stability. Finally, the single-NCM88 electrodes employing ZrO$_2$-LZCF cycled at 400 mA g$^{-1}$ and 30 °C showed a long cycle life of 90.8% capacity retention after 1000 cycles (Fig. 5j, k Supplementary Fig. 29d), which is well positioned in the state-of-the-art literature of ASSBs[7,20,25,26,31,38].

In summary, we reported a nanocomposite strategy for halide SEs to enhance their ionic conductivity and the compatibility with sulfide SEs. The mechanochemical reaction using Li$_2$O as an oxygen source created ZrO$_2$(-ACl)-A$_2$ZrCl$_6$ (A = Li or Na) with a network nanostructure wherein large-area interfaces were percolated. Despite the presence of the ionically insulating ZrO$_2$ phase in the HNSEs, interfacial superionic conduction enhanced ionic conductivities for Li$^+$ and Na$^+$ halide SEs: from 0.40 to 1.3 mS cm$^{-1}$ for ZrO$_2$-2LiCl-Li$_2$ZrCl$_6$ and from 0.011 to 0.11 mS cm$^{-1}$ for 0.13ZrO$_2$-0.61NaCl-0.26Na$_2$ZrCl$_6$. The applicability of the HNSE approach to other metals was highlighted, e.g., Al$_2$O$_3$-3Li$_2$ZrCl$_6$ (0.88 mS cm$^{-1}$), SnO$_2$-2Li$_2$ZrCl$_6$ (1.6 mS cm$^{-1}$) and 0.75ZrO$_2$-Li$_{2.25}$Zr$_{0.75}$Fe$_{0.25}$Cl$_6$ (1.4 mS cm$^{-1}$). DFT calculations combined with experimental synchrotron X-ray measurements and analysis revealed that the conduction behaviour at the ZrO$_2$/Li$_2$ZrCl$_6$ interface originated from local anion substitution, thereby leading to widened ion transport channels and increased local Li content at the interface. Partially oxidized Li$_2$ZrCl$_6$ at the populated interfaces was responsible for anomalous superionic conduction in the HNSEs, and active interfacial conduction was probed by $^6$Li MAS-NMR measurements and analysis. Our research work provides design principles for HNSEs, and they can be applied to other combinations of non-Li$^+$-conducting compounds and halide superionic conductors. In addition, the HNSE strategy counteracted the degradation of Li$^+$ conductivity by F-substitution. Finally, the HNSEs, especially the F-substituted HNSE ZrO$_2$-2Li$_2$ZrCl$_5$F, demonstrated good electrochemical energy storage performances in ASSB cells using LiCoO$_2$ or S-NCM88 positive electrodes and Li-In negative electrodes in terms of high-voltage stability up to 4.7 V (vs. Li/Li$^+$), compatibility with LPSCl and LiMO$_2$ cathode materials at 60 °C, rate capability, and long-term cycle life (82.0% capacity retention through 2000 cycles with respect to that at the 3rd cycle at 400 mA g$^{-1}$ and 30 °C). Notably, in this study, the HNSEs demonstrated the use of cost-effective elements, such as Zr and Al. Our approach provides not only an advancement in practical all-solid-state technology but also a dimension that widens the materials chemistry spaces for superionic conduction.

## Methods

### Preparation of materials

To prepare ZrO$_2$(-ACl)-A$_2$ZrCl$_6$ (A = Li or Na), a stoichiometric mixture of Li$_2$O (99.5%, Alfa Aesar) or Na$_2$O (80%, Sigma Aldrich, ~20% Na$_2$O$_2$), LiCl (99.99%, Sigma Aldrich) or NaCl (99.99%, Alfa Aesar), and ZrCl$_4$ (99.99%, Sigma Aldrich) was ball-milled at 600 rpm for 20 h in a ZrO$_2$ vial with ZrO$_2$ balls using Pulverisette 7PL (Fritsch GmbH) under Ar atmosphere. To prepare fluorinated HNSE ZrO$_2$-2Li$_2$ZrCl$_5$F, a stoichiometric mixture (Li$_2$O: ZrF$_4$: ZrCl$_4$ = 2: 0.5: 2.5) of Li$_2$O (99.5%, Alfa Aesar), ZrF$_4$ (99.9%, Sigma Aldrich) and ZrCl$_4$ (99.99%, Sigma Aldrich) was ball-milled under the same condition as for the conventional HNSEs. To prepare $n$M$_y$O$_z$-Li$_2$ZrCl$_6$ nanomixtures, M$_y$O$_z$ nanoparticles were ball-milled at 600 rpm for 20 h in a ZrO$_2$ vial with ZrO$_2$ balls using Pulverisette 7PL (Fritsch GmbH). ZrO$_2$ (99.95%, 20 nm) and Al$_2$O$_3$ (≥95%, 50 nm) nanopowders were purchased from Avention and Sigma Aldrich, respectively. Fumed SiO$_2$ powders were obtained from Sigma Aldrich. For the preparation of Li$_6$PS$_5$Cl, a stoichiometric mixture of Li$_2$S (99.9%, Alfa Aesar), P$_2$S$_5$ (99%, Sigma Aldrich), and LiCl (99.99%, Sigma Aldrich) was ball-milled at 600 rpm for 10 h in a ZrO$_2$ vial with ZrO$_2$ balls, followed by annealing at 550 °C for 6 h under an Ar atmosphere.

### Material characterization

Powder XRD patterns were collected using a Rigaku MiniFlex600 diffractometer with Cu K$_\alpha$ radiation ($\lambda = 1.5406$ Å). XRD cells containing hermetically sealed SE samples with a Be window were mounted on an XRD diffractometer and measured at 40 kV and 15 mA. X-ray total-scattering data were collected at beamline 28-ID-1 PDF at the National Synchrotron Light Source II (NSLSII) of Brookhaven National Laboratory with an X-ray energy of 74.5 keV ($\lambda = 0.1665$ Å). The prepared samples were loaded into polyimide (Kapton) tubes and hermetically sealed with epoxy resin. The 2D images were reduced to a 1D pattern with Ni calibrant using Dioptas software[57], and the PDF G(r) was obtained from Fourier transformation with a Q range of 1.5–23 Å$^{-1}$ from xPDFsuite[58]. The PDF G(r) of the HNSEs was refined using various structural models, including Li$_2$ZrCl$_6$ and ZrO$_2$ with adjustment of the scale factor, lattice parameter, and atomic displacement parameters. Zr K-edge XAS measurements were conducted at the 7D and 10 C beamlines of the Pohang Accelerator Laboratory (PAL) using a Si (111) double-crystal monochromator in transmission and fluorescence modes. Energy calibration was carried out using the reference spectra of the Zr metal foils. The Cl K-edge XANES spectra were measured in

fluorescence yield mode at the beamline 8-BM TES of NSLSII and 16A1 of Taiwan Light Source. XANES and EXAFS data were processed using the Demeter software package[59]. The extracted EXAFS signal, $k^3\chi(k)$, is Fourier transformed in the $k$-range of 3.2–11.2 Å$^{-1}$ and fitted in the R-range of 1.3–3.0 Å (Li$_2$ZrCl$_6$) and 1.3–3.2 Å (ZrO$_2$-2Li$_2$ZrCl$_6$ and nZrO$_2$-2Li$_2$ZrCl$_6$). $^6$Li MAS-NMR spectra were obtained at 170 K on a 400 MHz Advance II + system (Bruker solid-state NMR) at the KBSI Seoul Western Center, for which the $^6$Li resonance frequency was 58.862 MHz and a rotation frequency of 10 kHz was applied. The external chemical shift reference of the LiCl powder spun at 4 kHz was calibrated to 0 ppm. The NMR sample powders were sealed in a 4 mm ZrO$_2$ rotor in an Ar-filled glove box. $^6$Li$^+$-ion nonblocking symmetric cells of $^6$Li|LPSCl| (ZrO$_2$-2Li$_2$ZrCl$_6$)|LPSCl|$^6$Li were assembled as follows. $^6$Li foils were prepared by compressing $^6$Li chunks (95%, Sigma Aldrich). $^6$Li|LPSCl| (ZrO$_2$-2Li$_2$ZrCl$_6$)|LPSCl|$^6$Li cells were cycled at each cycle for 1 h at 500 µA cm$^{-2}$ and 60 °C. For the HRTEM measurements, the samples were loaded onto a lacey Cu grid, and HRTEM images were obtained using a JEM-ARM 200 F NEOARM (JEOL). For the TEM measurements in Supplementary Fig. 3g-i, the samples were loaded onto a lacey Cu grid and mounted on a double-tilt cryo-TEM holder with vacuum transfer (Double tilt LN2 Atmos Defend Holder, Mel-Build) to prevent air exposure of the samples. TEM images were obtained using a JEM-2100F (JEOL).

## Theoretical calculations

First-principles calculations were carried out using the Vienna Ab initio Simulation Package (VASP)[60]. Generalized gradient approximation (GGA) exchange-correlation with the Perdew–Burke–Ernzerhof (PBE) functional[61] was adopted, alongside the projector-augmented wave (PAW) method. A plane-wave cut-off energy of 520 eV was used, and the cell shape, cell volume, and atomic positions of each structure were fully relaxed until the forces on each atom were below 0.05 eV/Å.

Ab initio molecular dynamics (AIMD) simulations were performed to calculate Li$^+$ diffusivity and reveal its migration mechanism. These simulations used the NVT ensemble using a Nose–Hoover thermostat with a period of 80 fs[62]. The $1 \times 1 \times 2$ supercells of Li$_2$ZrCl$_6$ and other structures have lattice parameters larger than 10 Å in each direction, and a Γ-centred $1 \times 1 \times 1$ k-point grid was used. The heat-up process was executed for each supercell by raising the temperature from 100 K to each holding temperature (550–750 K) over 2 ps. AIMD simulations were conducted with a 2 fs interval time step for 200 ps at different holding temperatures, and diffusivities (D) were determined by linear fitting of the mean square displacement (MSD) of lithium ions using the following equations:

$$\text{MSD} = \frac{1}{N}\sum_{i=1}^{N}|r_i(t+\triangle t) - r_i(t)|^2 \qquad (1)$$

$$D = \frac{1}{2dtN}\sum_{i=1}^{N}|r_i(t+\triangle t) - r_i(t)|^2 \qquad (2)$$

where $r_i$ is the position of the i$^{th}$ ion at time t, Δt is the time step, N is the number of Li in the supercell structure, and d is the dimensionality factor. MSD and diffusivities were analysed by using the diffusion analyser module[63] in Pymatgen[64]. Diffusion channel size analysis based on the Li-ion trajectories generated from AIMD simulations was utilized using the topological analysis package Zeo$^{++}$[65].

Electrochemical stability windows were evaluated by constructing grand potential phase diagrams of all relevant phases with compositions of Li$_2$ZrCl$_6$, Li$_2$ZrCl$_5$F, Li$_2$ZrF$_6$ and Li$_3$Zr$_4$F$_{19}$ in equilibrium with the chemical potential of Li. The decomposition reaction energy is defined as follows:

$$\triangle E_D = E_{eq}(\text{Phase equlibria}, \mu_{Li}) - E_{SE}(\text{phase}) - \triangle n_{Li}\mu_{Li} \qquad (3)$$

where $\mu_{Li}$ is the chemical potential of Li and $\triangle n_{Li}$ is the number difference of elemental Li from the original composition. Interfacial chemical stability is calculated as the interface pseudobinary reaction energy between halide SE and contact material (cathode or sulfide SE), which is normalized by energy per formula unit of SE. The interface pseudobinary reaction energy is calculated as

$$\triangle E_{rxn}(C_{SE}, C_{CM}, x) = E_{eq}(xC_{SE} + (1-x)C_{CM}) - xE(C_{SE}) - (1-x)E(C_{CM}) \qquad (4)$$

where x is the molar fraction of the SE with $C_{SE}$ and $C_{CM}$ which are the compositions of halide SE and contact material (cathode or sulfide SE), respectively. For a given composition, the most stable convex hull energy (the lowest energy of the phase equilibrium) was used. The crystal structures of LiCoO$_2$ (space group: $R\bar{3}m$), Li$_6$PS$_5$Cl (space group: $F\bar{4}3m$) and all known compounds belonging to the Li-Zr-Cl-F systems were obtained from the Materials Project database[66], and their energies were calculated using the same DFT calculation parameters.

## Electrochemical characterization

Ionic conductivities were measured by the AC impedance method using ion-blocking Ti|SE|Ti symmetric cells. The cold-pressed pellets (40 mg, ≈500 µm) with a diameter of 6 mm were prepared at 370 MPa. The measurements were conducted 3 h after cell fabrication to ensure that thermal equilibrium was achieved. The EIS data were recorded for cells under an external pressure of ~70 MPa at open circuit voltage with an amplitude of 10 mV and a frequency range from 10 mHz to 7 MHz using a VSP-300 (Bio-Logic). Ten data points in each decade in frequency were recorded. For the Li-In||LiCoO$_2$ or LiNi$_{0.88}$Co$_{0.11}$Mn$_{0.01}$O$_2$ cells, Li-In was used as the counter and reference electrodes. After the Li-In powders with a nominal composition of Li$_{0.5}$In were prepared by ball-milling of In (Aldrich, 99%) and Li (FMC Lithium Corp.) at 2000 rpm, they were then mixed with Li$_6$PS$_5$Cl powders in a weight ratio of 8:2 at 2000 rpm. For the CV measurements, two kinds of SE powders (Li$_2$ZrCl$_6$ and ZrO$_2$-2Li$_2$ZrCl$_5$F) were manually mixed with super C65 with a weight ratio of 10:1. Li$_6$PS$_5$Cl powders (150 mg) were pelletized under 100 MPa to form SE layers. The SE-super C65 (Wellcos Co., Korea, BET surface area = 62 m$^2$ g$^{-1}$) mixture and Li-In electrodes were attached on either side of the SE layers, and the whole assembly was pressed at 370 MPa. The CV measurements were conducted using VMP3 (Bio-Logic) with a scan range from open-circuit voltage to 5 V (vs. Li/Li$^+$) at 0.1 mV s$^{-1}$. All-solid-state cells were fabricated as follows. For the preparation of the LPSCl monolayer, LPSCl powders (150 mg, ≈600 µm) were pelletized under 100 MPa. For the preparation of the bilayer, a thinner layer of ZrO$_2$-2Li$_2$ZrCl$_5$F (30 mg) was placed on the LPSCl layer and pelletized under 100 MPa. Composite working electrodes were prepared from a mixture of LiCoO$_2$ (Wellcos Co., Korea, D50 = 15.5 µm) or single-NCM88 (EcoPro BM, Korea, D50 = 3.3 µm), HNSEs, and super C65 powders with a weight ratio of 70:30:3 or 50:50:3. Finally, the LiCoO$_2$ or single-NCM88 electrodes (40-50 µm) and the Li-In electrodes (≈130 µm) were attached on either side of the SE layers, and the whole assembly was pressed at 370 MPa. The all-solid-state cells were tested under an external pressure of ~70 MPa. The specific current and specific capacity were determined based on the mass of active material, which was 10.2 mg for LCO and 7.3 mg for S-NCM88. The EIS measurements for the cells were performed from 1.5 MHz to 5 mHz at an amplitude of 10 mV after discharging the cells to 3.8 V (vs. Li/Li$^+$) at 16.4 mA g$^{-1}$ at the 2nd, 10th, and 100th cycles. To maintain a constant temperature during electrochemical tests, the cells were placed within an incubator at 30 °C. To ensure the reproducibility of the results, at least three cells were employed for each electrochemical test.

**Reporting summary**

Further information on research design is available in the Nature Portfolio Reporting Summary linked to this article.

## Data availability

The data generated or analysed in this study are provided in the paper and Supplementary Information and available from the corresponding author on reasonable request.

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

## Acknowledgements

This work was supported by Samsung Research Funding & Incubation Center of Samsung Electronics under project no. SRFC-MA2102-03 (D.-H.S., K.-W.N., and Y.S.J.) and by the National Research Foundation of Korea (NRF), funded by the Ministry of Science, ICT & Future Planning (2022M3J1A1085397 to Y.S.J.). The computational work was supported by the Supercomputing Center/Korea Institute of Science and Technology Information with supercomputing resources, including technical support (KSC-2022-CRE-0217 to D.-H.S.). The PDF and tender XAS research used beamline 28-ID–1(PDF) and 8-BM(TES) of the National Synchrotron Light Source II, a US Department of Energy (DOE) Office of Science User Facility operated for the DOE Office of Science by Brookhaven National Laboratory under contract no. DE-SC0012704 to G.K. Y.S.J. is grateful to Prof. Kyu Tae Lee at Seoul National University for a differential electrochemical mass spectrometry measurement.

## Author contributions

H.K. and Y.S.J. conceived the concept and designed the experiments. Y.S.J. supervised the work. H.K. and J.H.P. performed the syntheses and characterization of materials. J.K. carried out the electrochemical characterization. J.-S.K. and D.-H.S. performed the theoretical calculations. D.H., G.K., S.M.B., U.H., and K.-W.N. performed the synchrotron X-ray characterization. C.P., H.K., and H.-W.L. performed the cryo-TEM characterization. H.K. Y.S.J., D.-H.S., and K.-W.N. wrote the manuscript with discussion from all authors.

## Competing interests

The authors declare no competing interests.
