## [Peer Review File · Nature Communications]

REVIEWER COMMENTS

Reviewer #1 (Remarks to the Author):

Comments for Author

The authors describe a new halide nanocomposite solid electrolytes (HNSEs) $ZrO_2(-ACl)-A_2ZrCl_6$ ($A = Li$ or Na) serving as a high-quality interphase layer between cathodes and sulfide Li_6PS_5Cl (LPSCI) solid electrolyte, which promote the electrochemical performance of all-solid-state batteries (ASSBs). This work is of significance and novelty. The properties of solid electrolytes and the electrochemical performance of ASSBs are remarkable. The novel insight about “interfacial conduction enhancement” based on DFT calculation, synchrotron-based X-ray results, and MAS-NMR, is of importance to design superionic conductors. However, I have some concerns regarding the composition of materials and other weaknesses found in this paper. Overall, this work should be published in Nature Communications, subject to the following comments and questions:

Major Comments:

Materials synthesis and characterization

1. The authors claim the synthesis reaction mechanism: “ Li_2O oxidizes $ZrCl_4$ to form ZrO_2 nanoparticles...”(Line 109), “the oxidation of $ZrCl_4$ with Li_2O ”(Line 130), “The straightforward mechanochemical reaction using Li_2O as an oxidizer...”(Line 393).

However, I don't think this is an oxidation-reduction (redox) reaction since there are no changes in the oxidation state. The term “oxidizes” and “oxidizer” can not be used.

2. I have several concerns regarding the composition, purity, and homogeneity of the “ $ZrO_2-2Li_2ZrCl_6$ HNSE” sample. I doubt that there exist some residual nano-mixtures of $LiCl$, $ZrCl_4$, and Li_2O or other impurities in the ‘ $ZrO_2-2Li_2ZrCl_6$ HNSE’. Please provide more characterization and proof to identify the composition of ‘ $ZrO_2-2Li_2ZrCl_6$ ’.

(1) “The resulting HNSEs are thus comprised of Li_2ZrCl_6 , ZrO_2 , and $LiCl$, and their fractions are determined from the stoichiometric ratio of the precursors.”(Line 110-112)

The naming way of “ $ZrO_2-2Li_2ZrCl_6$ ” is often used to describe pure composite. However, I don't think the mechanochemical reaction of “ $ZrO_2-2Li_2ZrCl_6$ HNSE” is a high-homogeneity reaction. Therefore, the molecular formulas of “ $ZrO_2-2Li_2ZrCl_6$ HNSE” could not be simply “determined from the stoichiometric ratio of the precursors”.

(2) “The XRD signals of ZrO_2 were not observed, suggesting nanosized grains or poor crystallinity” (Line 121). I don't think the no observation of the XRD peak of ZrO_2 is due to nanosized grains, according to the Scherrer equation.

3. The authors claimed, "The fit of $ZrO_2-2Li_2ZrCl_6$ HNSE was unsuccessful, ending with an unacceptable R_w value of 48.3%, which suggested the necessity of including an additional (inter)phase to unambiguously describe its complicated structure." (Line 162-164)

Is possible that this complicated structure consists of $LiCl$, $ZrCl_4$, and Li_2O or other impurities rather than pure $ZrO_2-2Li_2ZrCl_6$?

4. The authors listed the chemical reaction equations of mechanochemical synthesis in Fig. 1(a) and the preparation protocols of HNSEs in Fig. S14.

The proposed chemical reaction equations in Fig. 1(a) need more references and discussions (or characterization, if possible) to be proved.

5. The authors claimed the significance of "fluorinated HNSE $ZrO_2-2Li_2ZrCl_5F$ ". However, the specific approach of fluorination is not detailed in the part of "Synthesis and characterization of HNSEs" in the manuscript, and only a simple preparation protocol is exhibited in Fig. S14. Please specify the details of fluorination in the manuscript.

Electrochemical characterization

6. How do different mass loadings of electrodes influence the electrochemical performance of ASSB full battery? Can ASSB full battery with high mass loading (mg/cm^2) deliver high capacity and stable cycling?

Please specify some key parameters of the battery: the loading (mg/cm^2) of cathode electrodes, the thickness of the solid electrolyte, and the thickness of the Li-In alloy anode if possible.

7. To ensure the accuracy of ionic conductivity results in Fig. 2 and Fig. 4, I suggest specifying the thickness and other essential parameters of each electrode determined in EIS experiments if possible.

8. Is there a problem with the volume change in the full battery? Why or why not?

9. Is there a problem with gas emissions in the full battery? Why or why not?

Theoretical calculations

10. In Fig. 3: How to determine the site of O substitution in the crystal of $Li_{2.5}ZrCl_{5.5}O_{0.5}$? How do the site of O substitution and O composition influence the channel size of $Li_{2.5}ZrCl_{5.5}O_{0.5}$?

11. In Fig. 4, the authors claimed that “Li₂ZrF₆ and Li₃Zr₄F₁₉ were thermodynamically stable passivating interphases that were decomposed from Li₂ZrCl₅F.” (Line 730-731)

(1) I am wondering whether the LiF can be decomposed from Li₂ZrCl₅F. Why or why not?

(2) Are there any similarities or differences between LiF, Li₂ZrF₆, and Li₃Zr₄F₁₉ as passivating interphases?

Minor Comments:

12. Please double-check the title of “Fig. 1 Synthesis and characterization of HNSEs (Li₂O-Li₂ZrCl₆)”. Is it “Li₂O-Li₂ZrCl₆” or “ZrO₂-Li₂ZrCl₆” (Line 700)?

13. “cost-effective and abundant elements, such as Zr and Al” (Line 415)

I don't think “Zr” is an “abundant” element although authors compared ZrCl₄ with other rare metal chlorides. There might be more appropriate words.

14. Please revise the wrong spell of ‘Thoertical calculations’ (Line 588)

Reviewer #2 (Remarks to the Author):

This paper reports interfacial superionic conduction in ZrO₂-Li₂ZrCl₆ system. This paper is interesting, but it needs to address the following questions:

(1) It is unclear that why nZrO₂-Li₂ZrCl₆ has lower conductivity, since Li₂ZrCl₆ should also have a percolation network.

(2) The author should acquire the NMR signal for ZrO₂-LZCF.

(3) Why does ZrO₂-LZCF perform better than ZrO₂-LZC? It is not well explained.

(4) What is the interaction between F and O in ZrO₂-LZCF system?

Reviewer #3 (Remarks to the Author):

The manuscript by Kwak targeted the preparation of novel halide nanocomposite

electrolyte compositions with general chemical formula $ZrO_2\text{-}AX\text{-}A_2ZrX_6$, with $A = \text{Li or Na}$, $X = \text{Cl, F}$, which are claimed to have good interfacial ionic conductivity as well as good stability against sulfide solid electrolytes. Furthermore, the author claim this new class of electrolytes appears stable against LiCoO_2 and high-voltage NMC electrodes under fast charging, relatively high-cycling temperatures (60 Celsius), and long-term cycle life. The mechanism of interfacial ion transport has been assessed using density functional theory. The validity of these mechanisms is further strengthened by magic-angle spinning solid-state NMR and specialized synchrotron experiments. Nicely, DFT could verify that oxygen “doping” on the Cl sites, increased the Zr-anion bond length, in agreement with the EXAFS analysis.

The manuscript is interesting and should be considered by Nature Communication after my comments are addressed.

1. The vendor and purity of the materials utilized in the synthesis of the halide nanocomposite should be reported.
2. In the result section, they wrote “the $ZrO_2\text{-}2Li_2ZrCl_6$ HNSE sample as it is a simple binary system and exhibited a much higher Li^+ conductivity of 1.1 mS cm^{-1} than Li_2ZrCl_6 (0.40 mS cm^{-1})” What is the conclusion here? Are they claiming that ZrO_2 in between grains modifies the texture of the grain boundary?
3. A Rietveld analysis of the XRD patterns in Figure 1b should be performed and added to this figure. Lattice constants, coordinates, fractional occupations, R factors, χ^2 , and CIF files should be provided. Presently, the authors just provide the lattice constants extracted from their PDF analysis.
4. They claimed that XANES is used to qualify Zr oxidation state, but I don’t see any K-edge of any standard compared there. Certainly, Li_2ZrCl_6 cannot be used as a standard material and something more established should be used instead.

5. High-resolution transmission electron microscopy is known to damage Li_2YCl_6 . I must admit that the micrographs in Figure S3 are not clear, and the diffraction not neat. I wonder whether Li_2ZrCl_6 derivatives can be equally damaged by the electron beam. The authors should clarify this.

6. Furthermore, the claim “Interestingly, the HRTEM images of the ZrO_2 - $2\text{Li}_2\text{ZrCl}_6$ HNSE (Figure 1f, g) showed that ZrO_2 formed a percolating network nanostructure with thickness of only a few nanometres.” Does not seem supported by the data. If they don’t have data backing this statement, this should be removed.

7. The statement “To the best of our knowledge, the Li^+ HNSE was the first inorganic superionic conductor that exploited the interfacial effect to promote conduction with ionic conductivity reaching 1 mS cm^{-1} .” appears a bit over the top. To me this is just the result of ZrO_2 modifying the texture of the microstructure, i.e. grain boundary, and hence improving the ion transport. These types of claims should be toned down.

8. Furthermore, it seems that many compounds reported in Tables S3 and S4 have a sizeable amount of both ZrO_2 and LiCl (or NaCl for the Na analogues) and are far from being phase pure. Now, this study clearly glosses over the importance of these impurities on the texture of grain boundaries, and thus the conductivity properties. A SEM analysis of the most important component is required. There may be a significant statistic of particle size which may also impact these observations.

9. Figure 2d, it’s unclear why there is just a datapoint for $n\text{ZrO}_2$ - $2\text{Li}_2\text{ZrCl}_6$. Why couldn’t the author measure the ion conductivity at lower or higher temperatures?

10. In the supplementary material, I don’t see any detail about the impedance analysis, starting from the equivalent circuits that have been used to extract the claimed ionic conductivities. This level of analysis is paramount. In particular, it’s not clear whether they’re able to deconvolute the grain vs bulk contribution to the total conductivity. Nice semi-circles are visible for the Na-based compounds. Have they tried?

11. The authors claimed that “ Li^+ diffusion was ~ 11 times faster for LZCO than LZC at 300 K ”. This is a very big claim considering that AIMD at a temperature as low as 300 cannot be considered converged. Perhaps is more valuable to make this comparison at higher temperatures.

Minor comments:

1. In the introduction, it should be “oxidative limits” not “oxidation limits”.

2. In the introduction, It’s true that oxide solid electrolytes are brittle, but this is not the main challenge. The main challenge in the utilization of oxide-based solid electrolytes is the cost-intensive process of sintering these materials, which is required at either very high temperatures or highly specialized systems such as spark plasma sintering.

RESPONSES TO REVIEWERS' COMMENTS

Dear Reviewers,

Thank you for your valuable comments. Our responses to each point brought up are given in blue.

Also, at the end, we listed up the changes made in the revised manuscript. The revised parts are shown in red in the revised manuscript. We believe that the paper is now stronger and thank the referees for their constructive comments very much.

Reviewer #1

The authors describe a new halide nanocomposite solid electrolytes (HNSEs) $ZrO_2(-ACl)-A_2ZrCl_6$ ($A = Li$ or Na) serving as a high-quality interphase layer between cathodes and sulfide Li_6PS_5Cl (LPSCl) solid electrolyte, which promote the electrochemical performance of all-solid-state batteries (ASSBs). This work is of significance and novelty. The properties of solid electrolytes and the electrochemical performance of ASSBs are remarkable. The novel insight about “interfacial conduction enhancement” based on DFT calculation, synchrotron-based X-ray results, and MAS-NMR, is of importance to design superionic conductors. However, I have some concerns regarding the composition of materials and other weaknesses found in this paper. Overall, this work should be published in Nature Communications, subject to the following comments and questions:

Response: We are thankful to the reviewer for a thorough review of our manuscript and the overall positive and constructive comments.

Major Comments:

Materials synthesis and characterization

Comment 1. *The authors claim the synthesis reaction mechanism: “ Li_2O oxidizes $ZrCl_4$ to form ZrO_2 nanoparticles...” (Line 109), “the oxidation of $ZrCl_4$ with Li_2O ”(Line 130), “The straightforward mechanochemical reaction using Li_2O as an oxidizer...”(Line 393). However, I don’t think this is an oxidation-reduction (redox) reaction since there are no changes in the oxidation state. The term “oxidizes” and “oxidizer” can not be used.*

Response to comment 1: We are thankful to the reviewer for the careful comment. In this manuscript, we used the term “oxidized” and “oxidizer” as the classical meaning of redox (the migration of oxygen atoms). According to the comment, the terms have been changed as the followings in the revised manuscript.

“ Li_2O oxidizes $ZrCl_4$ to form ZrO_2 nanoparticles,” → “ Li_2O reacts with $ZrCl_4$ to form ZrO_2 nanoparticles,”

“the oxidation of $ZrCl_4$ with Li_2O .” → “the reaction of $ZrCl_4$ with Li_2O .”

“The straightforward mechanochemical reaction using Li_2O as an oxidizer” → “The straightforward mechanochemical reaction using Li_2O as an oxygen source”

“Alternative oxidizing agents” → “Alternative oxygen sources”

Comment 2. *I have several concerns regarding the composition, purity, and homogeneity of the “ $ZrO_2-2Li_2ZrCl_6$ HNSE” sample. I doubt that there exist some residual nano-mixtures of $LiCl$, $ZrCl_4$, and Li_2O or other impurities in the ‘ $ZrO_2-2Li_2ZrCl_6$ HNSE’. Please provide more characterization and proof to identify the composition of ‘ $ZrO_2-2Li_2ZrCl_6$ ’.*

(1) *“The resulting HNSEs are thus comprised of Li_2ZrCl_6 , ZrO_2 , and $LiCl$, and their fractions are determined from the stoichiometric ratio of the precursors.”(Line 110-112)*

The naming way of “ $ZrO_2-2Li_2ZrCl_6$ ” is often used to describe pure composite. However, I don’t think the mechanochemical reaction of “ $ZrO_2-2Li_2ZrCl_6$ HNSE” is a high-homogeneity reaction. Therefore, the molecular formulas of “ $ZrO_2-2Li_2ZrCl_6$ HNSE” could not be simply “determined from the stoichiometric ratio of the precursors”.

- (2) “The XRD signals of ZrO₂ were not observed, suggesting nanosized grains or poor crystallinity” (Line 121). I don’t think the no observation of the XRD peak of ZrO₂ is due to nanosized grains, according to the Scherrer equation.

Response to comment 2: We are thankful to the reviewer for the insightful comments.

- (1) According to the comment, we have conducted a complementary analysis using DFT calculations, synchrotron XRD and PDF measurements. First, through DFT calculations, the chemical reaction energies are calculated using the energies of reactants and products (Supplementary Table 1). The reaction to generate ZrO₂ and LiCl from ZrCl₄ and Li₂O has a strong driving force (Equation S1, $\Delta E = -4.736$ eV). Furthermore, there is another spontaneous reaction from reactants (Li₂O, ZrCl₄, LiCl) to products (Li₂ZrCl₆, ZrO₂) when the molar ratios are stoichiometrically matched (Equation S2, $\Delta E = -5.000$ eV). Therefore, it is energetically not favorable for Li₂O to remain as an impurity after the reaction. Moreover, ternary-compound phase diagrams of ZrO₂-ZrCl₄-LiCl and Li-Zr-Cl are plotted in Supplementary Figure 2. No stable compounds exist except for Li₂ZrCl₆, indicating that the formation of impurity is energetically less favorable than one of Li₂ZrCl₆ and ZrO₂ although local off-stoichiometry may occur during the reaction.

Supplementary Table 1. Energy of reactants and products in the HNSE synthesis with the stoichiometric reaction equation of the in situ formation of ZrO₂.

Compound	Energy per formula unit (eV/f.u.)
LiCl	-7.436
ZrCl ₄	-24.825
Li ₂ O	-14.347
ZrO ₂	-28.511
Li ₂ ZrCl ₆	-39.763

$\text{LiCl} + 2\text{ZrCl}_4 + 2\text{Li}_2\text{O} \rightarrow 5\text{LiCl} + \text{ZrCl}_4 + \text{ZrO}_2$ ($\Delta E = -4.736$ eV) (Equation S1)
 $4\text{LiCl} + 5\text{ZrCl}_4 + 2\text{Li}_2\text{O} \rightarrow 4\text{Li}_2\text{ZrCl}_6 + \text{ZrO}_2$ ($\Delta E = -5.000$ eV) (Equation S2)

Supplementary Fig. 2 a,b, Ternary phase diagrams of ZrO₂-ZrCl₄-LiCl compound (a) and Li-Zr-Cl (b).

Furthermore, to verify the composition of ZrO₂-2Li₂ZrCl₆, synchrotron XRD and PDF measurements were carried out for a precursor mixture of Li₂O and ZrCl₄ (2:3 molar ratio) with varying ball-milling time (Supplementary Figures 3, and 4). The XRD patterns with increasing ball-milling time are classified into three regions (Supplementary Fig. 3). In region I corresponding to 2, 6, and 8 h, the peaks for LiCl emerge with slightly decreased peak intensities

for $ZrCl_4$ and Li_2O . In region II (10, 11, 12, and 16 h), the Li_2ZrCl_6 peaks evolve at the expense of the lowered peak intensities for $ZrCl_4$, Li_2O , and $LiCl$. Finally, after 20 h (region III), the XRD patterns remain almost identical with the exception of a minor decrease in the peak for Li_2O . Notably, for the sample ball-milled for 30 h, only the Li_2ZrCl_6 peaks are present with no detectable impurity or precursor peaks. Consistently, the PDF analysis results also show that, as the ball-mill time increases, the signal intensity for Li_2ZrCl_6 increases at the expense of the lowered intensities of the Li_2O and $ZrCl_4$ signals (Supplementary Figure 4). Moreover, the PDF analysis results confirm that the amount of Li_2ZrCl_6 does not increase further after 20 h. In addition, the evolution and increase of a ZrO_2 peak around 2 Å, corresponding to Zr-O bonding, is corroborated during the mechano-chemical milling.

In summary, the complementary analysis results unequivocally lead us to the following conclusions. The synthesis of ZrO_2 and Li_2ZrCl_6 from the reaction between Li_2O and $ZrCl_4$ is energetically favorable. The mechanochemical synthesis of ZrO_2 - $2Li_2ZrCl_6$ HNSE without a negligible amount of precursors or impurities is confirmed when enough time ($\geq \approx 20$ h) and energy are provided.

In the revised manuscript, Supplementary Figures 2, 3, and 4, and Supplementary Table 1 have been added. The relevant discussion has also been added in the Section of “Synthesis and characterization of HNSEs”. Also, the expression about the composition has been revised as follows.

“We extensively characterized the ZrO_2 - $2Li_2ZrCl_6$ HNSE sample as it is a simple binary system and exhibited a much higher Li^+ conductivity of 1.1 mS cm^{-1} than Li_2ZrCl_6 (0.40 mS cm^{-1}), despite the 7.86 vol.% of insulating ZrO_2 (based on the chemical formula of ZrO_2 - $2Li_2ZrCl_6$).”

Supplementary Fig. 3 Characterization of the synthesis reaction mechanism for HNSE by Synchrotron XRD. Synchrotron XRD patterns for the precursor mixture of Li_2O and ZrCl_4 (2:3 molar ratio) with varying ball-milling time.

Supplementary Fig. 4 Characterization of the synthesis reaction mechanism for HNSE by PDF. Synchrotron PDF $G(r)$ for the precursor mixture of Li_2O and ZrCl_4 (2:3 molar ratio) as a function of ball-milling time.

- (2) According to the HRTEM images, ZrO_2 in HNSEs exists as nanosized grains (~ 10 nm) with poor crystallinity (Figures 1f,g). Please read our response to the Reviewer 3's response 5 for the details. Besides, the PDF spectra reveal the ZrO_2 peaks up to ≈ 10 Å (Supplementary Figure 4), indicating that amorphous ZrO_2 is also present in HNSEs.^{R1} In conclusion, ZrO_2 in HNSEs exists as both crystalline and amorphous phases, while the latter occupies a larger fraction. The absence of ZrO_2 peaks in the XRD patterns (Figures 1b, and Supplementary Fig. 2) is thus understood.

In the revised manuscript, the term has been toned down as the following: “nanosized grain” “nanosized grain with poor crystallinity”.

R1 Zhang, F. *et al.* In situ Study of the Crystallization from Amorphous to Cubic Zirconium Oxide: Rietveld and Reverse Monte Carlo Analyses. *Chem. Mater.* **19**, 3118-3126, (2007).

Comment 3. The authors claimed, “The fit of ZrO_2 - $2\text{Li}_2\text{ZrCl}_6$ HNSE was unsuccessful, ending with an unacceptable R_w value of 48.3%, which suggested the necessity of including an additional (inter)phase to unambiguously describe its complicated structure.”(Line162-164)
Is possible that this complicated structure consists of LiCl , ZrCl_4 , and Li_2O or other impurities rather than pure ZrO_2 - $2\text{Li}_2\text{ZrCl}_6$?

Response to comment 3: We appreciate the reviewer pointing out the possibility of remaining

impurities in the final product. To verify the composition of the final product, we performed extensive PDF fitting refinements for $\text{ZrO}_2\text{-}2\text{Li}_2\text{ZrCl}_6$ (prepared by ball-milling for 20 h) using various combinations of structure models, including the LZCO interphase structure provided via DFT calculations, as indicated in the revised Supplementary Table 7. After such extensive preliminary refinements, the best-fit results are obtained using two different refinement ranges; low r range of 1.5–10 Å (region 1), high r range of 10–30 Å (region 2). In all cases, the medium and average structure of $\text{ZrO}_2\text{-}2\text{Li}_2\text{ZrCl}_6$ in the high r range (region 2) can be well represented by using a single Li_2ZrCl_6 structure (Supplementary Figure 17 and Table 10). For the low r range 1.5 ~ 10 Å (region 1), where the interface regime becomes prevailing, thus we have tested various combinations of model structures, including the precursors (e.g., LiCl and ZrCl_4), $\text{Li}_{2.5}\text{ZrCl}_{5.5}\text{O}_{0.5}$ (LZCO) interphase provided via DFT calculations, and Li_2O impurity, as tabulated in the supplementary Table 10. Remarkably, we achieved the excellent reliability factor and best-fit result when the significant amount of theoretically suggested LZCO interphase and minor Li_2O impurity are included (i.e., composition 3 in Supplementary Table 10). Please note that the reliability factor R_w of 48.3% in the case of using only two ZrO_2 and Li_2ZrCl_6 model structures in Fig. 1d is significantly lowered to 10.9% in the case of including the LZCO interphase. The calculated composition obtained by the PDF fit corresponds to $1.47\text{Li}_2\text{ZrCl}_2\text{-}0.36\text{Li}_{2.5}\text{ZrCl}_{5.5}\text{O}_{0.5}$ (interphase)- $1.01\text{ZrO}_2\text{-}0.16\text{Li}_2\text{O}$ (Supplementary Table 11). This result strongly supports the abundant LZCO interphase existing between ZrO_2 and LZC nanodomains.

We also performed the PDF fit for 30 h ball-milled $\text{ZrO}_2\text{-}2\text{Li}_2\text{ZrCl}_6$, which showed the best-fit result using the combinations of the same model structures as in the 20 h case. (Figure R1). Notably, the calculated composition of the 30 h sample showed significantly decreased impurity Li_2O amount (Table R1). Therefore, the Li_2O impurity could be eliminated from the final products if the mechanochemical reaction time is sufficient enough.

We also note that the inclusion of precursor LiCl and ZrCl_4 phases in the $\text{ZrO}_2\text{-}2\text{Li}_2\text{ZrCl}_6$ results in unreliable fitting results and increasing R_w during the PDF refinement, as shown in Supplementary Table 10. Therefore, we can unambiguously rule out the remaining precursors after sufficient ball-milling time for the $\text{ZrO}_2\text{-}2\text{Li}_2\text{ZrCl}_6$.

In the revised manuscript, Supplementary Figure 17 and Supplementary Tables 10 and 11 have been added. The relevant discussion has also been added in the “Interfacial Superionic Conduction of HNSEs” section.

Supplementary Fig. 17 Experimental PDF with best-fit results for $\text{ZrO}_2\text{-}2\text{Li}_2\text{ZrCl}_6$ in the 1.5–30 Å range. The PDF fitting was performed across different refinement ranges (low r range of 1.5–10 Å; high r range of 10–30 Å).

Supplementary Table 10 Reliable factor and mass fraction change obtained from PDF refinement in 1.5–10 Å with various compositions of precursors and products for $\text{ZrO}_2\text{-}2\text{Li}_2\text{ZrCl}_6$.

Model structure	Mass fraction (%)						R_w^a (%)
	$\text{Li}_6\text{Zr}_3\text{Cl}_{18}$	Zr_4O_8	$\text{Li}_{15}\text{Zr}_6\text{Cl}_{33}\text{O}_3$	Li_8O_4	Zr_2Cl_8	Li_4Cl_4	
Composition_1	82.6	17.4	-	-	-	-	14.6

Composition_2	54.5	13.3	32.2	-	-	-	12.3
Composition_3	53.9	14.3	31.3	0.55	-	-	10.9
Composition_4	83.6	15.4	-	0.91	-	-	13.9
Composition_5	80.9	14.6	-	-1.2	4.8	0.9	15.8
Composition_6	78.6	13.2	-	-4.0	8.5	-	12.5
Composition_7	89.2	17.2	-	-7.0	-	0.6	13.6

*a: Reliable factor

Supplementary Table 11 Composition of $ZrO_2-2Li_2ZrCl_6$ and $nZrO_2-2Li_2ZrCl_6$ calculated from the PDF refinement.

Sample	Composition ratio			
	Li_2ZrCl_6	ZrO_2	$Li_{2.5}ZrCl_{5.5}O_{0.5}$	Li_2O
$ZrO_2-2Li_2ZrCl_6$	1.47	1.01	0.36	0.16
$nZrO_2-2Li_2ZrCl_6$	2.05	0.95	-	-

Fig. R1 Experimental PDF fitted with multi-phase of Li_2ZrCl_6 , ZrO_2 , $\text{Li}_{2.5}\text{ZrCl}_{5.5}\text{O}_{0.5}$, and Li_2O for $\text{ZrO}_2\text{-}2\text{Li}_2\text{ZrCl}_6$ and 30 h ball-milled $\text{ZrO}_2\text{-}2\text{Li}_2\text{ZrCl}_6$ in the 1.5–30 Å range. The PDF fitting was performed across different refinement ranges (low r range of 1.5–10 Å; high r range of 10–30 Å)

Table R1 Composition ratio for $\text{ZrO}_2\text{-}2\text{Li}_2\text{ZrCl}_6$ and 30 h ball-milled $\text{ZrO}_2\text{-}2\text{Li}_2\text{ZrCl}_6$ calculated from PDF refinement.

Sample	Composition ratio			
	Li_2ZrCl_6	ZrO_2	$\text{Li}_{2.5}\text{ZrCl}_{5.5}\text{O}_{0.5}$	Li_2O
$\text{ZrO}_2\text{-}2\text{Li}_2\text{ZrCl}_6$	1.47	1.01	0.36	0.16
30 h ball-milled $\text{ZrO}_2\text{-}2\text{Li}_2\text{ZrCl}_6$	1.55	1.12	0.26	0.07

Comment 4. The authors listed the chemical reaction equations of mechanochemical synthesis in Fig. 1(a) and the preparation protocols of HNSEs in Fig. S14.

The proposed chemical reaction equations in Fig. 1(a) need more references and discussions (or characterization, if possible) to be proved.

Response to comment 4: We are thankful to the reviewer for the insightful comment. As discussed in our response to Comment 2, theoretically, the reaction of Li_2O with ZrCl_4 to form LiCl and ZrO_2 is spontaneous, which has also been verified experimentally by the XRD results shown in Figure R2. Furthermore, it was demonstrated that various HNSEs with different compositions, such as multimetal HNSEs, could be obtained by the multistep synthesis protocol: in the first step, Li_2O and metal chloride (MCl_y) react to form $\text{MO}_x\text{-LiCl}$ nanocomposites. The subsequent step involves the reaction with additional metal halides ($\text{M}'\text{X}_y$ (with $\text{M}''\text{X}_y$)) to form multimetal HNSEs, such as $\text{MO}_x\text{-LiCl-Li}_a\text{M}'\text{Cl}_b$ and $\text{MO}_x\text{-LiCl-Li}_a\text{M}''\text{Cl}_b$ (Supplementary Figure 21). Moreover, the synchrotron XRD and PDF results (Supplementary Figures 3 and 4) reveal that the reaction of Li_2O and ZrCl_4 proceeds by the evolution of ZrO_2 and LiCl and eventually the formation of Li_2ZrCl_6 . Please read our response to Comment 2.

In the revised manuscript, a discussion about the reaction mechanism has been added.

Fig. R2 Two-step preparation of $\text{ZrO}_2\text{-}2\text{Li}_2\text{ZrCl}_6$ HNSEs. XRD patterns for the product formed by the reaction of Li_2O and ZrCl_4 (top, step I) and the product formed by the subsequent reaction with ZrCl_4 (bottom, step II). The formation of LiCl and Li_2ZrCl_6 by step I and II is verified, respectively.

Comment 5. *The authors claimed the significance of “fluorinated HNSE ZrO₂-2Li₂ZrCl₅F”. However, the specific approach of fluorination is not detailed in the part of “Synthesis and characterization of HNSEs” in the manuscript, and only a simple preparation protocol is exhibited in Fig. S14. Please specify the details of fluorination in the manuscript.*

Response to comment 5: We are thankful to the reviewer for the careful comment. According to the comment, the synthesis procedure of the fluorinated HNSE ZrO₂-2Li₂ZrCl₅F has been added in the section of method (Preparation of materials) in the revised manuscript as the following.

“To prepare fluorinated HNSE ZrO₂-2Li₂ZrCl₅F, a stoichiometric mixture (Li₂O : ZrF₄ : ZrCl₄ = 2 : 0.5 : 2.5) of Li₂O (99.5%, Alfa Aesar), ZrF₄ (99.9%, Sigma Aldrich) and ZrCl₄ (99.99%, Sigma Aldrich) was ball-milled under the same condition as for the conventional HNSEs.”

Electrochemical characterization

Comment 6. *How do different mass loadings of electrodes influence the electrochemical performance of ASSB full battery? Can ASSB full battery with high mass loading (mg/cm²) deliver high capacity and stable cycling? Please specify some key parameters of the battery: the loading (mg/cm²) of cathode electrodes, the thickness of the solid electrolyte, and the thickness of the Li-In alloy anode if possible.*

Response to comment 6: We are thankful to the reviewer for the comment. According to the comment, all-solid-state cells with higher mass loadings (22.4 and 9.9 (mg of cathode material) cm⁻² for LCO and S-NCM88, respectively) were tested and the results are shown in Figure R3. The LCO electrode using ZrO₂-LZCF with the higher mass loading (20.4 mg cm⁻²) exhibited an outstanding cycling performance at 60 °C, which is consistent with the results for the lower mass loading (7.7 mg cm⁻²) (Figure R3a,b). However, the LCO electrode using ZrO₂-LZC showed a poorer capacity retention for the higher mass loading, compared with that for the lower mass loading. In addition, the rate capability results at 30 °C for the S-NCM88 electrode using ZrO₂-LZCF shows a significant degradation by increasing the mass loading from 3.7 to 9.9 mg cm⁻². Although the HNSE approach could counterbalance the degradation of ionic conductivity, the conductivity of ZrO₂-LZCF, 0.49 mS cm⁻¹, thus needs to be enhanced further for fulfilling the requirement of practical applications. Furthermore, engineering for composite electrodes such as dry coating of halide SEs could work to enhance the rate capability by maximizing ionic contacts with cathode active materials (CAMs).^{R2} These perspectives are included in our future research.

In the revised manuscript, the thickness (and mass loading) of the separating SE layers and Li-In counter electrodes has been added.

R2 Kim, J. S. *et al.* Synergistic halide-sulfide hybrid solid electrolytes for Ni-rich cathodes design guided by digital twin for all-solid-State Li batteries. *Energy Storage Mater.* **55**, 193-204, (2023).

Fig. R3 LCO and S-NCM88 ASSB cells with high mass loadings employing HNSEs (ZrO₂-2Li₂ZrCl₆ and ZrO₂-2Li₂ZrCl₅F). a, b, Cycling performances at 0.5C and 60 °C for LCO electrodes with a (ZrO₂-LZCF)/LPSCI bilayer cycled up to 4.3 V. c, Rate capabilities for S-NCM88 electrodes using ZrO₂-LZCF with different mass loadings of CAMs. LPSCI monolayers were used for the S-NCM88 cells.

Comment 7. To ensure the accuracy of ionic conductivity results in Fig. 2 and Fig. 4, I suggest specifying the thickness and other essential parameters of each electrode determined in EIS experiments if possible.

Response to comment 7: We are thankful to the reviewer for the comment. According to the comment, the thickness values of each pellet have been added to Supplementary Tables 5 and 6, and the discussion about the EIS measurements have been added in the Methods section. The equivalent circuit model and the fitted results are also provided in Supplementary Figure 10 and Supplementary Table 4.

Comment 8. Is there a problem with the volume change in the full battery? Why or why not?

Response to comment 8: We are thankful to the reviewer for the comment. As pointed out, the electrochemo-mechanical issues are critical for ASSBs.^{R3-7} For cathodes using halide SEs, side reactions between CAMs and SEs are marginal at mild conditions (e.g., cutoff voltage of 4.3 V and room temperature).^{R4,R8} Thus, the volume changes of CAMs during charge and discharge are the primary reason for the electrochemo-mechanical degradation in ASSBs. The disintegration of secondary particles of CAM and CAM/SE contact losses deteriorate Li⁺ and e⁻ transport pathways.^{R4-6} Especially, secondary particles of polycrystalline high-Ni NCM materials consisting of randomly oriented grains undergo severe disintegration in ASSB cells even at the initial cycles, leading to the poor initial Coulombic efficiency and fast capacity fading upon repeated cycling.^{R4-6} In our previous study, we demonstrated that, compared to polycrystalline CAMs, single-crystalline CAMs show better mechanical integrity and exhibited the excellent cycling stability after 100 cycles.^{R4} In this regard, to minimize the electrochemo-mechanical effects, the mechanochemically prepared HNSEs were applied for the single-crystalline CAMs of LiCoO₂ and NCA88 without any protective coating layers. In addition, a high operating pressure of 70 MPa helps suppress the electrochemo-mechanical degradation in ASSB cells. For practical application, an assessment under low pressures of a few megapascals at maximum and corresponding engineering, such as developments of zero-strain CAMs, advanced binders with improved mechanical strength, and highly deformable SEs, is imperative.^{R3,9-14} However, those are out of the scope in this study.

R3 Song, Y. *et al.* Electrochemo-mechanical effects as a critical design factor for all-solid-state batteries. *Curr. Opin. Solid State Mater. Sci.* **26**, 100977, (2022).

- R4 Han, Y. *et al.* Single- or Poly-Crystalline Ni-Rich Layered Cathode, Sulfide or Halide Solid Electrolyte: Which Will be the Winners for All-Solid-State Batteries? *Adv. Energy Mater.* **11**, 2100126, (2021).
- R5 Jung, S. H. *et al.* Ni-Rich Layered Cathode Materials with Electrochemo-Mechanically Compliant Microstructures for All-Solid-State Li Batteries. *Adv. Energy Mater.* **10**, 1903360, (2020).
- R6 Koerver, R. *et al.* Chemo-mechanical expansion of lithium electrode materials – on the route to mechanically optimized all-solid-state batteries. *Energy Environ. Sci.* **11**, 2142-2158, (2018).
- R7 Koerver, R. *et al.* Capacity Fade in Solid-State Batteries: Interphase Formation and Chemomechanical Processes in Nickel-Rich Layered Oxide Cathodes and Lithium Thiophosphate Solid Electrolytes. *Chem. Mater.* **29**, 5574-5582, (2017).
- R8 Kwak, H. *et al.* Emerging Halide Superionic Conductors for All-Solid-State Batteries: Design, Synthesis, and Practical Applications. *ACS Energy Lett.* **7**, 1776-1805, (2022).
- R9 Zhao, X. *et al.* Design principles for zero-strain Li-ion cathodes. *Joule* **6**, 1654-1671, (2022).
- R10 Ohzuku, T., Ueda, A. & Yamamoto, N. Zero-Strain Insertion Material of Li[Li_{1/3}Ti_{5/3}]O₄ for Rechargeable Lithium Cells. *J. Electrochem. Soc.* **142**, 1431, (1995).
- R11 Hippauf, F. *et al.* Overcoming binder limitations of sheet-type solid-state cathodes using a solvent-free dry-film approach. *Energy Storage Mater.* **21**, 390-398, (2019).
- R12 Kwon, T. Y. *et al.* Three-dimensional networking binders prepared in situ during wet-slurry process for all-solid-state batteries operating under low external pressure. *Energy Storage Mater.* **49**, 219-226, (2022).
- R13 Choi, S., Kwon, T.-w., Coskun, A. & Choi, J. W. Highly elastic binders integrating polyrotaxanes for silicon microparticle anodes in lithium ion batteries. *Science* **357**, 279-283, (2017).
- R14 Kato, A., Yamamoto, M., Sakuda, A., Hayashi, A. & Tatsumisago, M. Mechanical Properties of Li₂S–P₂S₅ Glasses with Lithium Halides and Application in All-Solid-State Batteries. *ACS Appl. Energy Mater.* **1**, 1002-1007, (2018).

Comment 9. *Is there a problem with gas emissions in the full battery? Why or why not?*

Response to comment 9: We are thankful to the reviewer for the comment. NCM cathode materials are known to cause gas evolution when used in liquid electrolyte-based cells.^{R15,16} In recent literature, an observation of the gas evolution in ASSB cells made of NCM and sulfide SE was reported.^{R17,18} The structure of NCM is unstable at a deeply delithiated state due to the formation of significant amounts of highly reactive Ni⁴⁺, which is prone to reduction to form the rock-salt-like phase on the surface.^{R5,16,19} The layered-to-rocksalt phase transformation process usually induces irreversible oxygen release from the surface of NCM.^{R19} Therefore, irreversible oxygen release can be prevented if the structural changes are suppressed.

In our previous result, microstructural evolution of NCM in ASSBs using sulfide or halide SEs was investigated.⁴⁰ In the NCM electrode employing sulfide SEs, after 100 cycles, a 20–30 nm thick surface layer that was distinct from the core region was observed. This layer was characterized as the NiO-like rocksalt structure. By contrast, NCA electrodes employing halide SEs did not show any difference in the crystal structure between the core and surface regions after 100 cycles. It is thus expected that there will be negligible gas evolution in ASSBs cells with NCA electrode using halide SEs.

During cycling, polycrystalline NCM shows significant particle fractures, which leads to contact loss and cell degradation. However, single-crystalline NCM does not suffer from noticeable cracking. According to a recent paper, polycrystalline NCM demonstrated significantly more oxygen release than single-crystalline NCM.^{R20} Because the carbonate content was very similar to the NCM employed in this paper and the cells were identical, this result indicates that the different cracking behavior (exposure of fresh and reactive surfaces) is coupled with the release of oxygen. To probe the gas evolution in situ in ASSB cells employing single-crystalline NCM88 with Li_2ZrCl_6 and $\text{ZrO}_2\text{-}2\text{Li}_2\text{ZrCl}_5\text{F}$, differential electrochemical mass spectrometry (DEMS) measurements were carried out for ASSB cells at 0.1C (Figure R4). During the operation of ASSB cells (first cycle between 3.0 – 4.3 V (vs. Li/Li^+) and second cycle between 3.0–4.8 V (vs. Li/Li^+)), no noticeable gas evolution was observed. The outstanding electrochemical oxidative stability of halide SE and the mechanical integrity of the single-crystalline feature of NCM88 suppress the phase transition of CAMs and thus inhibit oxygen release.

- R15 Jung, R., Metzger, M., Maglia, F., Stinner, C. & Gasteiger, H. A. Oxygen Release and Its Effect on the Cycling Stability of $\text{LiNi}_x\text{Mn}_y\text{Co}_z\text{O}_2$ (NMC) Cathode Materials for Li-Ion Batteries. *J. Electrochem. Soc.* **164**, A1361, (2017).
- R16 Ryu, H.-H., Park, K.-J., Yoon, C. S. & Sun, Y.-K. Capacity Fading of Ni-Rich $\text{Li}[\text{Ni}_x\text{Co}_y\text{Mn}_{1-x-y}]\text{O}_2$ ($0.6 \leq x \leq 0.95$) Cathodes for High-Energy-Density Lithium-Ion Batteries: Bulk or Surface Degradation? *Chem. Mater.* **30**, 1155-1163, (2018).
- R17 Bartsch, T. *et al.* Gas Evolution in All-Solid-State Battery Cells. *ACS Energy Lett.* **3**, 2539-2543, (2018).
- R18 Zuo, T.-T. *et al.* Impact of the Chlorination of Lithium Argyrodites on the Electrolyte/Cathode Interface in Solid-State Batteries. *Angew. Chem. Int. Ed.* DOI: 10.1002/anie.202213228.
- R19 Park, K.-J. *et al.* High-Capacity Concentration Gradient $\text{Li}[\text{Ni}_{0.865}\text{Co}_{0.120}\text{Al}_{0.015}]\text{O}_2$ Cathode for Lithium-Ion Batteries. *Adv. Energy Mater.* **8**, 1703612, (2018).
- R20 Payandeh, S. *et al.* The Effect of Single versus Polycrystalline Cathode Particles on All-Solid-State Battery Performance. *Adv. Mater. Interfaces* 2201806, (2022).

Fig. R4 Differential electrochemical mass spectrometry (DEMS) results of S-NCM88|Li-In ASSB cells. a, b, First two-cycle charge-discharge voltage profiles at 0.1C and 30 °C and corresponding time-resolved evolution rate for mass signals $m/z = 32$ (O_2), 44 ($^{12}CO_2$), and 70 (Cl_2) for using Li_2ZrCl_6 (a) and $ZrO_2-2Li_2ZrCl_5F$ (b).

Theoretical calculations

Comment 10. In Fig. 3: How to determine the site of O substitution in the crystal of $Li_{2.5}ZrCl_{5.5}O_{0.5}$? How do the site of O substitution and O composition influence the channel size of $Li_{2.5}ZrCl_{5.5}O_{0.5}$?

Response to comment 10: We are thankful to the reviewer for the comment. The oxygen-substituted Li_2ZrCl_6 ($Li_{12+x}Zr_6Cl_{36-x}O_x$ where $x = 1-4$) structures were prepared by enumeration technique. We considered all possible Cl and O orderings at Cl sites and selected 50 configurations with the lowest electrostatic energy. All the structures are fully relaxed with DFT calculation. We checked the channel size of the most stable structure at each composition (Figure R5a), and the optimal point which has the largest channel size and the highest lattice volume is expected to be $x \approx 3$ in $Li_{12+x}Zr_6Cl_{36-x}O_x$ ($Li_{15}Zr_6Cl_{33}O_3$). Those are mainly attributed to the lattice expansion effects by oxygen and the lattice contraction effects by lithium. Moreover, as our EXAFS data show the increased bond length of Zr-Cl in $ZrO_2-2Li_2ZrCl_6$ HNSE (Figure 1c), we compared the average bond length and radial distribution functions (RDFs) of Zr-Cl in the $Li_{12+x}Zr_6Cl_{36-x}O_x$ structures. The average bond length of Zr-Cl increases significantly at $x = 3$ or more, which can be also confirmed by the RDF results (Figure R5b).

Among the generated structures of $Li_{15}Zr_6Cl_{33}O_3$ ($Li_{2.5}ZrCl_{5.5}O_{0.5}$), we selected the structures in which oxygen is placed at one side (since oxygen substitution mainly occurs at the interface of Li_2ZrCl_6 and ZrO_2 and chemical potential difference drives the degree of oxygen substitution gradient). For 20

structures, we conducted short AIMD screening during 20 ps at 700 K with topological analysis and the characteristics of structures that show rapid diffusion were analyzed (Figure R6). As the repulsion between adjacent $\text{ZrCl}_{6-x}\text{O}_x$ polyhedra increases, the ion conduction channel widens, enabling rapid Li^+ diffusion. Here, when the oxygen ions of neighboring $\text{ZrCl}_{6-x}\text{O}_x$ polyhedra face each other (agglomerated) in the c-axis direction, such effects are maximized, resulting in fast Li^+ diffusion. Furthermore, the energy difference between the most stable $\text{Li}_{2.5}\text{ZrCl}_{5.5}\text{O}_{0.5}$ structure (O-dispersed) and one with such characteristics (O-agglomerated, Figure 3a) is only 9.4 meV/atom. Thus, oxygen-agglomerated structures can be sufficiently formed when anion exchange occurs at the interfaces of $\text{ZrO}_2\text{-Li}_2\text{ZrCl}_6$.

In the revised manuscript, Supplementary Figure 13 and Supplementary texts have been added.

Fig. R5 Degree of oxygen substitution in Li_2ZrCl_6 by composition. a,b Channel size (a) and simulated radial distribution function (RDF) with average bond lengths of Zr-Cl (b) in $\text{Li}_{12+x}\text{Zr}_6\text{Cl}_{36-x}\text{O}_x$ ($x = 1\sim 4$).

Fig. R6 Topological analysis with MSD value of $\text{Li}_{2.5}\text{ZrCl}_{5.5}\text{O}_{0.5}$. a, Crystal structures and MSD values of $\text{Li}_{2.5}\text{ZrCl}_{5.5}\text{O}_{0.5}$. b, Topological analysis and channel size of $\text{Li}_{2.5}\text{ZrCl}_{5.5}\text{O}_{0.5}$. c, Short AIMD screening during 20 ps at 700 K for crystal structure E.

Supplementary Fig. S13 Screening process of oxygen-substituted Li_2ZrCl_6 (LZC) and $\text{Li}_{2.5}\text{ZrCl}_{5.5}\text{O}_{0.5}$. **a**, Topological analysis and lattice volume of the most stable oxygen-substituted structures of LZC ($\text{Li}_{12}\text{Zr}_6\text{Cl}_{36} \rightarrow \text{Li}_{12+x}\text{Zr}_6\text{Cl}_{36-x}\text{O}_x$, $x = 1\sim 4$) for determining O-substituted composition. **b-c**, Several screened structures among screened 50 structures (**b**) in the composition of $\text{Li}_{1.5}\text{Zr}_6\text{Cl}_{33}\text{O}_3$ ($= \text{Li}_{2.5}\text{ZrCl}_{5.5}\text{O}_{0.5}$) with mean square displacement (MSD) value obtained by short AIMD screening (700 K, 20 ps) and Li transport channel size of each structure (**c**). **d**, Short AIMD screening during 20 ps at 700 K for crystal structure E.

Comment 11. In Fig. 4, the authors claimed that “ Li_2ZrF_6 and $\text{Li}_3\text{Zr}_4\text{F}_{19}$ were thermodynamically stable passivating interphases that were decomposed from $\text{Li}_2\text{ZrCl}_5\text{F}$.” (Line 730-731)

- (1) I am wondering whether the LiF can be decomposed from $\text{Li}_2\text{ZrCl}_5\text{F}$. Why or why not?
- (2) Are there any similarities or differences between LiF , Li_2ZrF_6 , and $\text{Li}_3\text{Zr}_4\text{F}_{19}$ as passivating interphases?

Response to comment 11: We are thankful to the reviewer for the insightful comment.

- (1) Our calculations show that $\text{Li}_2\text{ZrCl}_5\text{F}$ (LZCF) can be decomposed into the LiF phase at low voltage (Li insertion) as suggested in Supplementary Table 14. However, since LZCF is used as a catholyte in our system that operates down to 3.0 V (vs. Li/Li^+), it is not expected to form LiF phase.
- (2) We summarize their similarities and differences as following; Similarities: By the influence of negative ions (fluorine), those lithium fluoride compounds have higher oxidation stability than other halide compounds such as chlorides and oxides.²¹ Our DFT calculations also show that they

have very high band gaps (6.31, 5.63, and 8.65 eV for Li_2ZrF_6 , $\text{Li}_3\text{Zr}_4\text{F}_{19}$, and LiF , respectively), suggesting that they can be a passivating phase (electronic insulator and Li^+ conductor) for cathode materials (Figure R7). Differences: When $\text{Li}_2\text{ZrCl}_5\text{F}$ is exposed to high-voltage conditions, it will be sequentially decomposed into Li_2ZrF_6 and $\text{Li}_3\text{Zr}_4\text{F}_{19}$ phases, whereas LiF can be formed under very low-voltage conditions. Li-Zr-F ternary compounds (6.526 and 6.613 V for Li_2ZrF_6 and $\text{Li}_3\text{Zr}_4\text{F}_{19}$, respectively) tend to be more stable than LiF (6.332 V) at high voltage (Figure R7). LiF can also be formed as a reduction-limited decomposition product of Li-Zr-F ternary compounds, and it has a much lower oxidation voltage limit (0 V) than Li_2ZrF_6 (1.239 V) and $\text{Li}_3\text{Zr}_4\text{F}_{19}$ (1.506 V).

Fig. R7 Band gap, intrinsic stability window, and decomposed phase equilibria of Li_2ZrF_6 (a), $\text{Li}_3\text{Zr}_4\text{F}_{19}$ (b) and LiF (c).

Minor Comments:

Comment 12. Please double-check the title of “Fig. 1 Synthesis and characterization of HNSEs ($\text{Li}_2\text{O-Li}_2\text{ZrCl}_6$)”. Is it “ $\text{Li}_2\text{O-Li}_2\text{ZrCl}_6$ ” or “ $\text{ZrO}_2\text{-Li}_2\text{ZrCl}_6$ ” (Line 700)?

Response to comment 12: We are thankful to the reviewer for the kind comment. According to the comment, the chemical formula has been revised as “ $\text{ZrO}_2\text{-2Li}_2\text{ZrCl}_6$ ” in the revised manuscript.

Comment 13. “cost-effective and abundant elements, such as Zr and Al” (Line 415) I don’t think “Zr” is an “abundant” element although authors compared ZrCl_4 with other rare metal chlorides. There might be more appropriate words.

Response to comment 13: We are thankful to the reviewer for the comment. According to the reviewer’s comment, the expression has been toned down in the revised manuscript.

Comment 14. Please revise the wrong spell of ‘Theoretical calculations’ (Line 588)

Response to comment 14: We are thankful to the reviewer for the comment. In the revised manuscript, the sentence has been revised.

Reviewer #2

This paper reports interfacial superionic conduction in ZrO₂-Li₂ZrCl₆ system. This paper is interesting, but it needs to address the following questions:

Response: We are thankful to the reviewer for a thorough review of our manuscript and the overall positive and constructive comments.

Comment 1. *It is unclear that why nZrO₂-Li₂ZrCl₆ has lower conductivity, since Li₂ZrCl₆ should also have a percolation network.*

Response to comment 1: We are thankful to the reviewer for the comment. The ZrO₂-2Li₂ZrCl₆ HNSE showed a roughly threefold improvement in Li⁺ conductivity compared with that of Li₂ZrCl₆ (1.1 vs. 0.40 mS cm⁻¹ at 30 °C). Although it was substantially lower than that of the HNSE ZrO₂-2Li₂ZrCl₆, the Li⁺ conductivity of nZrO₂-2Li₂ZrCl₆ was also higher (0.60 mS cm⁻¹) than that of Li₂ZrCl₆. This result indicates a promoted Li⁺ conduction at the interfaces between ZrO₂ and Li₂ZrCl₆. However, the sizes of ZrO₂ nanoparticles in nZrO₂-2Li₂ZrCl₆ are 20-50 nm and substantially larger than those for HNSEs (only a few nanometres), reflecting much lower interfacial area between ZrO₂ and Li₂ZrCl₆ for nZrO₂-2Li₂ZrCl₆ than for HNSEs. The Li⁺ conductivity of nZrO₂-2Li₂ZrCl₆, which is higher than that of Li₂ZrCl₆ but lower than that of HNSEs, is thus rationalized.

Comment 2. *The author should acquire the NMR signal for ZrO₂-LZCF.*

Response to comment 2: We are thankful to the reviewer for the comment. As discussed in the manuscript, the shoulder peak observed in the ZrO₂-2Li₂ZrCl₆ HNSE spectrum corresponds to the O-substituted phase suggested by the DFT calculations. According to the comment, ⁶Li MAS-NMR spectrum of ZrO₂-2Li₂ZrCl₅F was obtained and is shown in Supplementary Figure 25. Similar to the spectrum of the ZrO₂-2Li₂ZrCl₆ HNSE, ZrO₂-2Li₂ZrCl₅F also exhibited the shoulder peak indicated by an arrow, suggesting an O-substituted Li₂ZrCl₅F interphase. This interphase likely promotes interfacial superionic conduction and boosts Li⁺ conduction despite fluorination.

Supplementary Fig. 25 ⁶Li MAS-NMR spectrum for ZrO₂-2Li₂ZrCl₅F. Spectra for Li₂ZrCl₆ and ZrO₂-2Li₂ZrCl₆ are also compared.

Comment 3. Why does ZrO₂-LZCF perform better than ZrO₂-LZC? It is not well explained.

Response to comment 3: We are thankful to the reviewer for the insightful comment.

For the cells tested at an elevated temperature of 60 °C, when the LPSCl monolayer was used so that LPSCl was in direct contact with cathodes, the cathodes using ZrO₂-LZCF significantly outperform those using LZC (Figure 5b,c). In contrast, when the LZCF|LPSCl bilayer was used so that LPSCl was not exposed to high voltages, the capacity degradation for using LZC was substantially alleviated but still significant. These results suggest two origins of halide SEs for the degradation of the cathodes using LZC at elevated temperature: i) Incompatibility with LPSCl at high voltages. ii) High-voltage instability.

First, to assess the former, the DFT calculations and control EIS experiments were carried out. The mutual reaction energies of the halide/LPSCl SE mixtures were over 0.4 eV/f.u., strongly indicating their poor compatibility (Supplementary Table 16). For both LZC and ZrO₂-LZCF, the EIS experiment using Ti(halide-LPSCl mixture)|Ti cells stored at 60 °C showed marginal changes in Nyquist plots after a week (Supplementary Figure 31), indicating a stable halide-LPSCl interface under no electrochemical driving forces. A control EIS experiment was also performed using (halide-C)|LPSCl|(Li-In) cells charged to 4.3 V (vs. Li/Li⁺) at 60 °C so that the halide-LPSCl interfaces are subjected to the high voltage (Supplementary Figure 32). The result exhibited the continuously increased impedance for the cells using LZC. In contrast, the impedance increased for an hour and then saturated for the cell comprised of the ZrO₂-LZCF HNSE. This result indicates excellent passivating behaviour for using ZrO₂-LZCF. From these results, it can be concluded that the halide/sulfide incompatibility is driven electrochemically at elevated temperatures. Importantly, F-substituted chloride SEs are free from this halide/sulfide incompatibility issue.

Second, to assess the high-voltage stability, the CV measurements and DFT calculations were carried out. In the course of preparing the revision, reliable CV results for (halide/C)|halide |LPSCl|(Li-In) cells at 0.1 mV s⁻¹ and 30 °C were obtained and are shown in Figure 4c and Supplementary Table 13. ZrO₂-2Li₂ZrCl₅F exhibited a remarkably smaller integrated anodic current of 1.98 mA V g⁻¹ up to 5.0 V (vs. Li/Li⁺), compared to Li₂ZrCl₆ (2.76 mA V g⁻¹). The difference became even larger at the second cycle (0.55 vs. 2.00 mA V g⁻¹ for ZrO₂-2Li₂ZrCl₅F and Li₂ZrCl₆, respectively). Furthermore, Li₂ZrCl₆ showed a cathodic peak at ≈3.5 V (vs. Li/Li⁺) at the first cycle and they intensified further at the second cycle, which is indicated by an asterisk. It is postulated that the cathodic currents originate from decomposition byproducts formed during the prior positive scan.⁵³ By contrast, ZrO₂-2Li₂ZrCl₅F exhibited negligible cathodic currents, strongly suggesting its excellent passivating behavior. The DFT results consistently revealed that the oxidative limit of Li₂ZrCl₅F (4.274 V vs. Li/Li⁺) was slightly lower than that of Li₂ZrCl₆ (4.307 V vs. Li/Li⁺), but the formation of desirable F-based passivating interphase materials such as Li₂ZrF₆ and Li₃Zr₄F₁₉ can increase the range of the anodic limit (Figure 4d and Supplementary Table 14).^{37,38}

In conclusion, compared with LZC, the fluorinated HNSE ZrO₂-LZCF exhibits remarkably improved sulfide (LPSCl) compatibility and high-voltage stability. These explain the drastic enhancement in cycling performance at the elevated temperature for the LCO electrodes by applying ZrO₂-LZCF, compared with those for using LZC.

In the revised manuscript, the relevant discussion has been revised. A reference 53 has also been

added.

- 53 Shao, Q. *et al.* New Insights into the Effects of Zr Substitution and Carbon Additive on $\text{Li}_{3-x}\text{Er}_{1-x}\text{Zr}_x\text{Cl}_6$ Halide Solid Electrolytes. *ACS Appl. Mater. Interfaces* **14**, 8095-8105, (2022).

Fig. 4c, First- and second-cycle CV curves for (SE-C)|SE|LPSCI|(Li-In) cells between 3.0 and 5.0 V (vs. Li/Li⁺) at 0.1 mV s⁻¹ and 30 °C.

Comment 4. *What is the interaction between F and O in ZrO₂-LZCF system?*

Response to comment 4: We are thankful to the reviewer for the insightful comment. To investigate the interaction between anions (O/Cl and O/F) in ZrO₂-LZC and ZrO₂-LZCF systems, we built a slab model of ZrO₂(010)-Li₂ZrCl_{5,x}F_{1-x}(001) and calculated DFT energies with and without anionic exchange in the interface (Figure R8). The spontaneous driving force (-0.408 eV) for oxygen-chlorine exchange is confirmed as we have proved experimentally and computationally. In the same way, it is also checked whether the exchange between fluorine from LZCF and oxygen from ZrO₂ is energetically favorable or not when fluorine is present at the surface. Since oxygen-fluorine exchange has a stronger driving force (-0.534 eV), it is speculated that oxygen-fluorine exchange can occur in ZrO₂-LZCF interfaces.

Fig. R8 Interfacial slab model of $\text{ZrO}_2(010)\text{-Li}_2\text{ZrCl}_{5-x}\text{F}_{1-x}(001)$ and driving force for anion exchange at the interface

Reviewer #3

The manuscript by Kwak targeted the preparation of novel halide nanocomposite electrolyte compositions with general chemical formula $ZrO_2-AX-A_2ZrX_6$, with $A = Li$ or Na , $X = Cl, F$, which are claimed to have good interfacial ionic conductivity as well as good stability against sulfide solid electrolytes. Furthermore, the author claim this new class of electrolytes appears stable against $LiCoO_2$ and high-voltage NMC electrodes under fast charging, relatively high-cycling temperatures (60 Celsius), and long-term cycle life. The mechanism of interfacial ion transport has been assessed using density functional theory. The validity of these mechanisms is further strengthened by magic-angle spinning solid-state NMR and specialized synchrotron experiments. Nicely, DFT could verify that oxygen “doping” on the Cl sites, increased the Zr-anion bond length, in agreement with the EXAFS analysis. The manuscript is interesting and should be considered by Nature Communication after my comments are addressed.

Response: We are thankful to the reviewer for a thorough review of our manuscript and the overall positive and constructive comments.

Major Comments:

Comment 1. *The vendor and purity of the materials utilized in the synthesis of the halide nanocomposite should be reported.*

Response to comment 1: We are thankful to the reviewer for the comment. As shown in the Method section, the information on the materials utilized has been reported.

Comment 2. *In the result section, they wrote “the $ZrO_2-2Li_2ZrCl_6$ HNSE sample as it is a simple binary system and exhibited a much higher Li^+ conductivity of 1.1 mS cm^{-1} than Li_2ZrCl_6 (0.40 mS cm^{-1})” What is the conclusion here? Are they claiming that ZrO_2 in between grains modifies the texture of the grain boundary?*

Response to comment 2: We are thankful to the reviewer for the comment. The presence of insulating materials in SEs is expected to be disadvantageous. Despite 7.86 vol.% of insulating ZrO_2 (based on the theoretical formula), the $ZrO_2-2Li_2ZrCl_6$ HNSE sample showed a much higher Li^+ conductivity of 1.1 mS cm^{-1} than that of Li_2ZrCl_6 (0.40 mS cm^{-1}). We revealed that the anomalous enhancement in Li^+ conductivity originates from the oxygen-substituted Li_2ZrCl_6 interphase at the populated interfaces in HNSEs, which was suggested by the DFT calculations results and supported by the synchrotron X-ray and 6Li MAS-NMR results.

Comment 3. *A Rietveld analysis of the XRD patterns in Figure 1b should be performed and added to this figure. Lattice constants, coordinates, fractional occupations, R factors, χ^2 , and CIF files should be provided. Presently, the authors just provide the lattice constants extracted from their PDF analysis.*

Response to comment 3: We are thankful to the reviewer for the careful comment. We agree with the reviewer that detailed structure information should be provided. However, Rietveld refinement can only be performed for the XRD data with sufficient peak resolution and intensity. Although we measured XRD in synchrotron ($\lambda = 0.1665\text{ \AA}$), XRD peaks for the ZrO_2 phase in $ZrO_2-2Li_2ZrCl_6$ were not observable. In addition, the Li_2ZrCl_6 peaks featured a broad shape and weak intensity due to low

crystallinity and nanosize (revised Supplementary Figure 3). Therefore, even synchrotron XRD data was insufficient to perform Rietveld refinement, which is why we have adopted pair distribution function (PDF) analysis. PDF is a powerful method for quantitatively analyzing the nanosized and disordered materials since both Bragg and diffuse scattering are measured. The PDF refinements for Li_2ZrCl_6 , $\text{ZrO}_2\text{-}2\text{Li}_2\text{ZrCl}_6$, and $n\text{ZrO}_2\text{-}2\text{Li}_2\text{ZrCl}_6$ were performed using a multi-phase model, and the lattice parameters with a reliable factor are tabulated in the revised Supplementary Table 9. Also, we attach CIF files obtained from PDF refinement for providing structural information.

In the revised manuscript, Supplementary Table 9 has been modified as follows.

Supplementary Table 9 Lattice parameters for Li_2ZrCl_6 , $\text{ZrO}_2\text{-}2\text{Li}_2\text{ZrCl}_6$, and $n\text{ZrO}_2\text{-}2\text{Li}_2\text{ZrCl}_6$ calculated from PDF fitting analysis.

Sample	R_w^a (%)	Lattice parameter									
		$\text{Li}_2\text{ZrCl}_6^d$		ZrO_2^e				$\text{Li}_{2.5}\text{ZrCl}_{5.5}\text{O}_{0.5}^f$			Li_2O^g
		a	c	a	b	c	β	a	b	c	a
Li_2ZrCl_6	17.0	10.97	5.92	-				-			-
ZrO_2	11.6			5.14	5.20	5.31	99.24				-
$\text{ZrO}_2\text{-}2\text{Li}_2\text{ZrCl}_6$	10.9 ^b 16.4 ^c	10.96	5.91	5.48	5.83	5.00	103.7	10.97	9.72	15.44	4.79
$n\text{ZrO}_2\text{-}2\text{Li}_2\text{ZrCl}_6$	13.8	10.96	5.93	5.13	5.21	5.33	99.63	-			-

* a: Reliable factor, b: Reliable factor fitted in 1.5–10 Å, c: Reliable factor fitted in 10–30 Å

* d: $P\bar{3}m1$ (SG# 164), e: $P2/c$ (SG# 13), f: $P1$ (Modeled structure), g: $Fm\bar{3}m$ (SG# 225)

Comment 4. They claimed that XANES is used to qualify Zr oxidation state, but I don't see any K-edge of any standard compared there. Certainly, Li_2ZrCl_6 cannot be used as a standard material and something more established should be used instead.

Response to comment 4: We are thankful to the reviewer for the comment. As the reviewer's suggestion, we have added Zr K-edge XANES spectra of ZrCl_4 and ZrO_2 references in Supplementary Figure 5. The bond strength differs with ligand elements and thus can affect absorption edge energy. As Zr-Cl bonding is weaker than Zr-O bonding, the absorption edge of ZrO_2 is positioned at slightly higher energy than the ZrCl_4 spectrum. A comparison of the edge position with the reference spectra confirms that the average oxidation state of Zr is tetravalent (4+) in the $\text{ZrO}_2\text{-}2\text{Li}_2\text{ZrCl}_6$ and $n\text{ZrO}_2\text{-}2\text{Li}_2\text{ZrCl}_6$ samples.

Supplementary Fig. 5 XANES results. a,b Zr K-edge (a) and Cl K-edge (b) XANES spectra of Li_2ZrCl_6 , $\text{ZrO}_2\text{-}2\text{Li}_2\text{ZrCl}_6$, and $n\text{ZrO}_2\text{-}2\text{Li}_2\text{ZrCl}_6$. The reference samples spectra for ZrCl_4 and ZrO_2 nanopowders ($n\text{ZrO}_2$) are also compared in (a).

Comment 5. High-resolution transmission electron microscopy is known to damage Li_2YCl_6 . I must admit that the micrographs in Figure S3 are not clear, and the diffraction not neat. I wonder whether Li_2ZrCl_6 derivatives can be equally damaged by the electron beam. The authors should clarify this.

Response to comment 5: We are thankful to the reviewer for the helpful comment. As pointed out, vulnerable halide SE materials are damaged by electron beams when conventional TEM measurements are performed. In this regard, cryogenic TEM (cryo-TEM), which emerged as a powerful tool for the analysis of electron-beam-sensitive materials, such as biomolecules, Li metal, and solid electrolyte interphases, should be highly promising for halide SE materials. Accordingly, cryo-HRTEM measurements of $n\text{ZrO}_2\text{-}2\text{Li}_2\text{ZrCl}_6$ and $\text{ZrO}_2\text{-}2\text{Li}_2\text{ZrCl}_6$ were conducted at cryogenic temperature ($-175\text{ }^\circ\text{C}$) without any exposure to ambient air using a double-tilt cryogenic TEM holder. Clear images and corresponding diffraction patterns were acquired by cryo-TEM measurements, from which local nanostructures are visualized (Figures 1e-g). For $n\text{ZrO}_2\text{-}2\text{Li}_2\text{ZrCl}_6$, crystalline ZrO_2 nanoparticles with diameters ranging from 20 to 30 nm are embedded in glass-ceramic Li_2ZrCl_6 matrices (Figure 1e and Supplementary Figure 6). The cryo-HRTEM images of the $\text{ZrO}_2\text{-}2\text{Li}_2\text{ZrCl}_6$ HNSE show that poor-crystalline nanograins ($< 20\text{ nm}$) are randomly dispersed in glass-ceramic Li_2ZrCl_6 matrices (Figures 1f,g and Supplementary Figure 6). This observation confirms the much larger-area percolating interfaces for $\text{ZrO}_2\text{-}2\text{Li}_2\text{ZrCl}_6$ compared with $n\text{ZrO}_2\text{-}2\text{Li}_2\text{ZrCl}_6$.

In the revised manuscript, the cryo-HRTEM images and corresponding fast Fourier transform (FFT) images have been added in Figure 1e-g and Supplementary Figure 6.

Fig. 1 e-g, Cryo-HRTEM images of $n\text{ZrO}_2-2\text{Li}_2\text{ZrCl}_6$ (e) and $\text{ZrO}_2-2\text{Li}_2\text{ZrCl}_6$ at lower magnification (f) and higher magnification (g). The red and yellow outlines indicate the domains of ZrO_2 and Li_2ZrCl_6 , respectively. Schematics of the nanostructures are shown in the insets of (e) and (f).

Supplementary Fig. 6 Cryo-HRTEM results of Li^+ HNSEs. **a-d**, Cryo-HRTEM image of $n\text{ZrO}_2-2\text{Li}_2\text{ZrCl}_6$ (**a**) and corresponding FFT patterns (**b**). **c,d**, Inverse FFT images from the circled region of the FFT pattern in the inset image, corresponding to Li_2ZrCl_6 (**c**) and ZrO_2 (**d**). **e-h**, Cryo-HRTEM image of $\text{ZrO}_2-2\text{Li}_2\text{ZrCl}_6$ (**e**) and corresponding FFT patterns (**f**). **g, h** Inverse FFT images from the circled region of the FFT pattern in the inset image, corresponding to Li_2ZrCl_6 (**g**) and ZrO_2 (**h**).

Comment 6. Furthermore, the claim “Interestingly, the HRTEM images of the ZrO₂-2Li₂ZrCl₆ HNSE (Figure 1f, g) showed that ZrO₂ formed a percolating network nanostructure with thickness of only a few nanometres.” Does not seem supported by the data. If they don't have data backing this statement, this should be removed.

Response to comment 6: We thank the reviewer for the careful comment. As responded in Comment 5, the cyro-HRTEM images with clear diffraction patterns support that nanograin domains of ZrO₂ with sizes < 20 nm are dispersed in the glass-ceramic Li₂ZrCl₆ matrix and ZrO₂ formed a local percolating network nanostructure. However, the thickness of the network nanostructure ranges from 10 to 20 nm and it is unclear whether ZrO₂ nanodomains form a universal percolating network inside the Li₂ZrCl₆ matrix.

In the revised manuscript, the relevant description has been revised.

Comment 7. The statement “To the best of our knowledge, the Li⁺ HNSE was the first inorganic superionic conductor that exploited the interfacial effect to promote conduction with ionic conductivity reaching 1 mS cm⁻¹.” appears a bit over the top. To me this is just the result of ZrO₂ modifying the texture of the microstructure, i.e. grain boundary, and hence improving the ion transport. These types of claims should be toned down.

Response to comment 7: We are thankful to the reviewer for the comment. The interfacial conduction effect was first reported by Liang et al. in 1930.⁴² Specifically, the Li⁺ conductivity of LiI was improved from 10⁻⁷ to 10⁻⁵ S cm⁻¹ by the addition of metal oxides such as Al₂O₃. Since then, for various heterostructured materials, anomalous enhancements in ionic conductivity have been reported.^{42,43,44,45,46,48} The examples include LiF/silica films (6 × 10⁻⁶ S cm⁻¹ at 50 °C),⁴⁵ LiBH₄-LiI/Al₂O₃, (1 × 10⁻⁴ S cm⁻¹ at 20 °C),⁴⁸ and PAN with LiClO₄/Li_{0.33}La_{0.557}TiO₃ (LLTO) nanowires (6.1 × 10⁻⁵ S cm⁻¹ at 30 °C).⁴⁶ However, none of these SEs showed ionic conductivities exceeding 1 mS cm⁻¹ at room temperature. To promote the interfacial ionic conductivity effect, a large interfacial area between two different materials is imperative. In most previous literature, the simple mixing of two different materials has been a common practice. This protocol has limitations in terms of downsizing and the high cost of nano-fillers. In this work, we have demonstrated that ZrO₂ nanograins with only a few nanometers could be generated in situ by the straightforward mechanochemical method, which is also unprecedented and advantageous in terms of cost. The higher Li⁺ conductivity of the ZrO₂-Li₂ZrCl₆ HNSE (1.3 mS cm⁻¹) than that of *n*ZrO₂-Li₂ZrCl₆ prepared by mixing ZrO₂ nanoparticles with Li₂ZrCl₆ (0.60 mS cm⁻¹) highlights the novelty of our strategy. Also, considering that conventional compositional tuning, such as ion substitution, has been common to enhance the ionic conductivity of halide SEs, our approach unlocks a new dimension for superionic conduction in materials chemistry. In previous literature regarding interfacial conduction, underlying mechanisms have remained unclear, although various effects, such as the space charge layer and crystallinity effects, were proposed.^{42, 43, 45, 47} In our study, we have revealed that the interfacial superionic conduction originates from the oxygen-substituted compounds at the interface. Finally, we demonstrated that our strategy is not restricted to specific compositions. The HNSEs could be formed for Na analogues (0.04 mS cm⁻¹ for ZrO₂-Na₂ZrCl₆, and 0.11 mS cm⁻¹ for 0.13ZrO₂-0.61NaCl-0.26Na₂ZrCl₆) and multi-metal HNSEs such as Al₂O₃-3Li₂ZrCl₆ and SnO₂-2Li₂ZrCl₆, which show even higher Li⁺ conductivities (max. 1.6 mS cm⁻¹ for SnO₂-2Li₂ZrCl₆) than ZrO₂-2Li₂ZrCl₆. In summary, our strategy is the first case that achieves the ionic conductivity of 1 mS cm⁻¹ by the interfacial superionic conduction effect. It is also universal and expandable. Moreover, we believe that the elucidation of the mechanism is of

significance to the design of superionic conductors.

Comment 8. Furthermore, it seems that many compounds reported in Tables S3 and S4 have a sizeable amount of both ZrO₂ and LiCl (or NaCl for the Na analogues) and are far from being phase pure. Now, this study clearly glosses over the importance of these impurities on the texture of grain boundaries, and thus the conductivity properties. A SEM analysis of the most important component is required. There may be a significant statistic of particle size which may also impact these observations.

Response to comment 8: We are thankful to the reviewer for the comment. Based on the DFT calculation results in Supplementary Figure 2, Li₂ZrCl₆ is the only stable phase in the ZrO₂-ZrCl₄-LiCl ternary region. Therefore, it is possible to synthesize various HNSEs including LiCl or ZrO₂ with possible off-stoichiometry. At the Li₂ZrCl₆/ZrO₂ interface, O-substituted Li₂ZrCl₆ with high Li⁺ conductivity is formed. The DFT calculation, NMR, and PDF analysis results support the formation and interfacial superionic conduction of the O-substituted Li₂ZrCl₆ in ZrO₂-2Li₂ZrCl₆. Interestingly, 0.53LiCl-Li₂ZrCl₆ showed higher ionic conductivities of 0.70 mS cm⁻¹ than that for Li₂ZrCl₆ (0.40 mS cm⁻¹). Even though this work focused on the ZrO₂/Li₂ZrCl₆ interfaces, the enhanced ionic conduction at the LiCl/Li₂ZrCl₆ interfaces is of course an intriguing issue which could further be investigated. Generally, highly crystalline SE domains can be identified at the grain and grain boundary at the micrometer scale by SEM measurement. However, mechano-chemically prepared HNSEs include poorly crystalline and nanosized grains of ZrO₂ and Li₂ZrCl₆. Thus, the identification of the grain and grain boundary of ZrO₂ and Li₂ZrCl₆ by SEM is not possible due to the limited resolution of SEM. Therefore, we have carried out the cryo-HRTEM analysis for HNSEs. The cryo-HRTEM analysis results of ZrO₂-2Li₂ZrCl₆ demonstrate that poor-crystalline nanograins of ZrO₂ are spread randomly in the glass-ceramic Li₂ZrCl₆ matrix (Figures 1e-g and Supplementary Figure 6). The significantly larger-area percolating interfaces for ZrO₂-2Li₂ZrCl₆ compared with *n*ZrO₂-2Li₂ZrCl₆ are confirmed by this observation. Please read our response to the Comment 5 for the detail. Nevertheless, according to the reviewer's comment, we obtained SEM images for Li₂ZrCl₆, ZrO₂-2Li₂ZrCl₆, and ZrO₂-2Li₂ZrCl₆F (Figure R9). The samples showed irregular particles with sizes at the micrometer scale without distinct differences from each other.

Fig. R9 SEM images of halide SEs and HNSEs. a-c, SEM images of Li₂ZrCl₆ (a), ZrO₂-2Li₂ZrCl₆ (b) and ZrO₂-2Li₂ZrCl₆F (c).

Comment 9. Figure 2d, it's unclear why there is just a datapoint for *n*ZrO₂-2Li₂ZrCl₆. Why couldn't the author measure the ion conductivity at lower or higher temperatures?

Response to comment 9: We are thankful to the reviewer for the comment. According to the comment, ionic conductivities of *n*ZrO₂-2Li₂ZrCl₆ were obtained at different temperatures (Figure

2c). In the whole temperature range, Li^+ conductivities of $n\text{ZrO}_2\text{-}2\text{Li}_2\text{ZrCl}_6$ were lower than those of HNSE ($\text{ZrO}_2\text{-}2\text{Li}_2\text{ZrCl}_6$) but higher than those of LZC.

In the revised manuscript, the corresponding data have been added in Figure 2c.

Fig. 2 c, Arrhenius plots of ionic conductivities for conventional halide SEs and HNSEs.

Comment 10. *In the supplementary material, I don't see any detail about the impedance analysis, starting from the equivalent circuits that have been used to extract the claimed ionic conductivities. This level of analysis is paramount. In particular, it's not clear whether they're able to deconvolute the grain vs bulk contribution to the total conductivity. Nice semi-circles are visible for the Na-based compounds. Have they tried?*

Response to comment 10: We are thankful to the reviewer for the insightful comment. All the EIS data for the conductivity measurements were fitted using an equivalent circuit model shown in Figure R10a. The equivalent circuit model and the fitted results are also provided in **Supplementary Figure 10** and **Supplementary Table 4**. As pointed out, in Nyquist plots of SE materials, signals for grain (or bulk) and grain-boundary resistances evolve at higher and lower frequencies, respectively, and could often be deconvoluted especially at low temperatures.^{R21-23} In SEs, the grain and grain boundary capacitance values usually lie in $\approx 10^{-12}$ F and 10^{-11} - 10^{-8} F, respectively.^{R22,23} However, the EIS data for ionic conductivity in the original manuscript do not show two distinct semicircles or asymmetric features. Therefore, an EIS measurement for $\text{ZrO}_2\text{-}2\text{Li}_2\text{ZrCl}_6$ was conducted at a lower temperature of -40 °C. The corresponding Nyquist plot is presented in Figure R11. Unfortunately, it also showed one semicircle with decent symmetry. Nevertheless, the data for $\text{ZrO}_2\text{-}2\text{Li}_2\text{ZrCl}_6$ were fitted with an equivalent circuit model representing the grain and grain boundary contributions as shown in Figure R10b. However, it fails to deconvolute the contributions. Instead, the data were fitted well with the equivalent circuit model with the one parallel-connected RC component (Figure R10a). And the capacitance value for the single semicircle is 27.71 pF, which corresponds to the bulk (grain)

transport. Therefore, the contribution for total resistances for HNSEs by grain boundary resistances is likely marginal.

- R21 Bron, P., Dehnen, S. & Roling, B. $\text{Li}_{10}\text{Si}_{0.3}\text{Sn}_{0.7}\text{P}_2\text{S}_{12}$ – A low-cost and low-grain-boundary-resistance lithium superionic conductor. *J. Power Sources* **329**, 530-535, (2016).
- R22 Irvine, J. T. S., Sinclair, D. C. & West, A. R. Electroceramics: Characterization by Impedance Spectroscopy. *Adv. Mater.* **2**, 132-138, (1990).
- R23 Kraft, M. A. *et al.* Inducing High Ionic Conductivity in the Lithium Superionic Argyrodites $\text{Li}_{6+x}\text{P}_{1-x}\text{Ge}_x\text{S}_5\text{I}$ for All-Solid-State Batteries. *J. Am. Chem. Soc.* **140**, 16330-16339, (2018).

Supplementary Table 4 Fitted results of the EIS data shown in Figures 2, 4, and Supplementary Figures 12, 23, and 24. Corresponding equivalent circuit model is presented in Supplementary Figure 10.

HNSE	Thickness (um)	R_1 (Ω)	C (pF)	α	Conductivity (mS cm^{-1})	Corresponding Figure no.
Li_2ZrCl_6	640	555.2	39.72	0.922	0.4	
$\text{ZrO}_2\text{-}2\text{Li}_2\text{ZrCl}_6$	540	168.1	11.58	0.904	1.13	Fig. 2a
$n\text{ZrO}_2\text{-}2\text{Li}_2\text{ZrCl}_6$	510	303.8	-	-	0.6	
Na_2ZrCl_6	600	18536	53.03	0.914	0.011	
$\text{ZrO}_2\text{-}2\text{Na}_2\text{ZrCl}_6$	540	3380	84.57	0.908	0.057	Fig. 2b & Supplementary Fig.12
$0.13\text{ZrO}_2\text{-}0.61\text{NaCl}\text{-}0.26\text{Na}_2\text{ZrCl}_6$	480	1576	100.1	0.897	0.11	
$\text{Li}_{2.25}\text{Zr}_{0.75}\text{Fe}_{0.25}\text{Cl}_6$	570	209.5	6.69	0.907	0.96	
$0.9\text{ZrO}_2\text{-}2\text{Li}_{2.1}\text{Zr}_{0.9}\text{Fe}_{0.1}\text{Cl}_6$	520	153.3	0.28	0.944	1.20	
$0.75\text{ZrO}_2\text{-}2\text{Li}_{2.25}\text{Zr}_{0.75}\text{Fe}_{0.25}\text{Cl}_6$	500	126.1	-	-	1.40	Supplementary Fig. 23
$0.6\text{ZrO}_2\text{-}2\text{Li}_{2.4}\text{Zr}_{0.6}\text{Fe}_{0.4}\text{Cl}_6$	520	182	2.59	0.906	1.01	
$\text{Li}_2\text{ZrCl}_5\text{F}$	610	622.6	22.26	0.921	0.35	
$\text{Li}_2\text{ZrCl}_{4.5}\text{F}_{1.5}$	640	790.6	27.94	0.899	0.29	Fig. 4b & Supplementary Fig. 24
$\text{Li}_2\text{ZrCl}_4\text{F}_2$	610	904.7	31.74	0.914	0.24	
$\text{ZrO}_2\text{-}2\text{Li}_2\text{ZrCl}_5\text{F}$	520	377.6	19.7	1	0.49	

Fig. R10 Equivalent circuit schemes.

Fig. R11 Nyquist plot of ion-blocking Ti|SE|Ti symmetric cells for $\text{ZrO}_2\text{-}2\text{Li}_2\text{ZrCl}_6$ at $-40\text{ }^\circ\text{C}$.

Comment 11. The authors claimed that “ Li^+ diffusion was ~ 11 times faster for LZCO than LZC at 300 K”. This is a very big claim considering that AIMD at a temperature as low as 300 cannot be considered converged. Perhaps is more valuable to make this comparison at higher temperatures.

Response to comment 11: As the reviewer knows, an extremely long computational time for AIMD simulations is required to predict statistically reliable Li^+ diffusivity at room temperature (300 K) due to slow Li^+ hopping at low temperatures. Thus, it is technically impossible to directly predict the diffusivity of material with a conductivity of about 1 mS cm^{-1} at 300 K using AIMD. This is why Li^+ diffusivity at 300 K has been predicted by extrapolating a few values of Li^+ diffusivity calculated above 500 K, and we also followed the general method.^{R24-26} However, we agree with the reviewer that this is a very big claim as we assume that the diffusivity follows the Arrhenius relationship without the slope change from high to low temperatures. As the reviewer suggested, we will present diffusivity data at high temperatures as an additional supplementary table to prove not only that it is faster for LZCO than LZC at high temperatures but also that the slope is less steep for LZCO, which reflects higher diffusion at 300 K. In addition, the statements have been revised to avoid misunderstanding of the results.

In the revised manuscript, Supplementary Table 12 has been added, and the relevant discussion has been revised as follows.

- R24 Deng, Z., Zhu, Z., Chu, I.-H. & Ong, S. P. Data-Driven First-Principles Methods for the Study and Design of Alkali Superionic Conductors. *Chem. Mater.* **29**, 281-288, (2017).
- R25 Patel, S. V. *et al.* Tunable Lithium-Ion Transport in Mixed-Halide Argyrodites $\text{Li}_{6-x}\text{PS}_{5-x}\text{ClBr}_x$: An Unusual Compositional Space. *Chem. Mater.* **33**, 1435-1443, (2021).
- R26 Banerjee, S., Holekevi Chandrappa, M. L. & Ong, S. P. Role of Critical Oxygen Concentration in the $\beta\text{-Li}_3\text{PS}_{4-x}\text{O}_x$ Solid Electrolyte. *ACS Appl. Energy Mater.* **5**, 35-41, (2022).

“ Li^+ diffusion was ~11 times faster for LZCO than LZC at 300 K.” → “LZCO shows a faster diffusion than LZC at all temperatures for which AIMD simulations were performed, but also has a gentle slope compared with the diffusivities of LZC. It is expected that Li^+ diffusion shows ~11 times faster for LZCO than LZC at 300 K.”

Supplementary Table 12. The diffusivity corresponding to each temperature obtained through AIMD calculations.

Temperature (K)	Diffusivity of Li^+ ($\text{cm}^2 \text{s}^{-1}$)	
	LZC	LZCO
550	1.05×10^{-5}	1.73×10^{-5}
600	1.34×10^{-5}	2.94×10^{-5}
700	3.09×10^{-5}	3.27×10^{-5}
725	3.64×10^{-5}	4.29×10^{-5}
750	4.08×10^{-5}	5.34×10^{-5}
300 (Extrapolated)	8.92×10^{-8}	9.88×10^{-7}

Minor Comments:

Comment 12. In the introduction, it should be “oxidative limits” not “oxidation limits”.

Response to comment 12: We are thankful to the reviewer for the comment. In the revised manuscript, the term has been revised.

Comment 13. In the introduction, It’s true that oxide solid electrolytes are brittle, but this is not the main challenge. The main challenge in the utilization of oxide-based solid electrolytes is the cost-intensive process of sintering these materials, which is required at either very high temperatures or highly specialized systems such as spark plasma sintering.

Response to comment 13: We are thankful to the reviewer for the careful comment. According to the comment, we have revised the relevant sentence and added references 18 and 19.

- 18 Kim, S. *et al.* High-energy and durable lithium metal batteries using garnet-type solid electrolytes with tailored lithium-metal compatibility. *Nat. Commun.* **13**, 1883, (2022).
- 19 Kim, K. H. *et al.* Characterization of the interface between LiCoO_2 and $\text{Li}_7\text{La}_3\text{Zr}_2\text{O}_{12}$ in an all-solid-state rechargeable lithium battery. *J. Power Sources* **196**, 764-767, (2011).

REVIEWERS' COMMENTS

Reviewer #1 (Remarks to the Author):

This work is of significance and novelty. The authors describe a new halide nanocomposite solid electrolytes (HNSEs) $\text{ZrO}_2(-\text{AlCl})-\text{A}_2\text{ZrCl}_6$ (A = Li or Na) serving as a high-quality interphase layer between cathodes and sulfide $\text{Li}_6\text{PS}_5\text{Cl}$ (LPSCI) solid electrolyte, which promote the electrochemical performance of all-solid-state batteries (ASSBs). The properties of solid electrolytes and the electrochemical performance of ASSBs are remarkable. The corresponding mechanism analysis by DFT calculation, synchrotron-based X-ray results, and MAS-NMR is convincing. The novel finding about “interfacial conduction enhancement” provides new insight into designing superionic conductors for ASSBs.

The authors have comprehensively responded to my fourteen comments with solid details. I have no more comments.

Overall, this work could be considered published in Nature Communications.

Reviewer #2 (Remarks to the Author):

It is satisfactory now.

Reviewer #3 (Remarks to the Author):

I read in depth the amendments that the authors made to the manuscript. I support the publication of this work in Nature Communications.

RESPONSES TO REVIEWERS' COMMENTS

Dear Reviewers,

Thank you for your valuable comments. Our responses are given in blue. We believe that the paper is now stronger and thank the referees for their constructive comments very much.

Reviewer #1

Comment: *This work is of significance and novelty. The authors describe a new halide nanocomposite solid electrolytes (HNSEs) $ZrO_2(-ACl)-A_2ZrCl_6$ ($A = Li$ or Na) serving as a high-quality interphase layer between cathodes and sulfide Li_6PS_5Cl (LPSCl) solid electrolyte, which promote the electrochemical performance of all-solid-state batteries (ASSBs). The properties of solid electrolytes and the electrochemical performance of ASSBs are remarkable. The corresponding mechanism analysis by DFT calculation, synchrotron-based X-ray results, and MAS-NMR is convincing. The novel finding about “interfacial conduction enhancement” provides new insight into designing superionic conductors for ASSBs.*

The authors have comprehensively responded to my fourteen comments with solid details. I have no more comments.

Overall, this work could be considered published in Nature Communications.

Response: Thank you for taking the time to review our manuscript. We appreciate your thorough review and appreciate your positive and constructive comments. We believe that the paper is now stronger and thank the referees for their valuable comments very much.

Reviewer #2

It is satisfactory now.

Response: We are thankful to the reviewer for your positive comment.

Reviewer #3

I read in depth the amendments that the authors made to the manuscript. I support the publication of this work in Nature Communications.

Response: We are thankful to the reviewer for your positive comment.